# DFML: Decentralized Federated Mutual Learning

**Yasser H. Khalil**                                    *yasser.khalil1@huawei.com*
*Huawei Noah's Ark Lab, Montreal, Canada.*

**Amir H. Estiri**                              *amir.hossein.estiri1@huawei.com*
*Huawei Noah's Ark Lab, Montreal, Canada.*

**Mahdi Beitollahi**                              *mahdi.beitollahi@huawei.com*
*Huawei Noah's Ark Lab, Montreal, Canada.*

**Nader Asadi**                                         *nader.asadi@huawei.com*
*Huawei Noah's Ark Lab, Montreal, Canada.*

**Sobhan Hemati**                                    *sobhan.hemati@huawei.com*
*Huawei Noah's Ark Lab, Montreal, Canada.*

**Xu Li**                                                      *xu.lica@huawei.com*
*Huawei Technologies Canada Inc., Ottawa, Canada.*

**Guojun Zhang**                                     *guojun.zhang@huawei.com*
*Huawei Noah's Ark Lab, Montreal, Canada.*

**Xi Chen**                                               *xi.chen4@huawei.com*
*Huawei Noah's Ark Lab, Montreal, Canada.*

**Reviewed on OpenReview:** *https: // openreview. net/ forum? id= I9HvzJbUbh*

## Abstract

In the realm of real-world devices, centralized servers in Federated Learning (FL) present challenges including communication bottlenecks and susceptibility to a single point of failure. Additionally, contemporary devices inherently exhibit model and data heterogeneity. Existing work lacks a Decentralized FL (DFL) framework capable of accommodating such heterogeneity without imposing architectural restrictions or assuming the availability of additional data. To address these issues, we propose a **D**ecentralized **F**ederated **M**utual **L**earning (DFML) framework that is serverless, supports nonrestrictive heterogeneous models, and avoids reliance on additional data. DFML effectively handles model and data heterogeneity through mutual learning, which distills knowledge between clients, and cyclically varying the amount of supervision and distillation signals. Extensive experimental results demonstrate consistent effectiveness of DFML in both convergence speed and global accuracy, outperforming prevalent baselines under various conditions. For example, with the CIFAR-100 dataset and 50 clients, DFML achieves a substantial increase of **+17.20%** and **+19.95%** in global accuracy under Independent and Identically Distributed (IID) and non-IID data shifts, respectively.

## 1 Introduction

Federated Learning (FL) stands as a promising paradigm in machine learning that enables decentralized learning without sharing raw data, thereby enhancing data privacy. Although, Centralized FL (CFL) has been predominant in the literature (McMahan et al., 2017; Alam et al., 2022; Diao et al., 2020; Horvath et al., 2021; Caldas et al., 2018), it relies on a central server. Communication with a server can be a bottleneck,

especially when numerous dispersed devices exist, and a server is vulnerable to a single point of failure. To avoid these challenges, Decentralized FL (DFL) serves as an alternative, facilitating knowledge sharing among clients without the need of a central server (Beltrán et al., 2023; Yuan et al., 2023; Giuseppi et al., 2022; Li et al., 2021). DFL also offers computational and energy efficiency, as resources are distributed across clients instead of being concentrated in a centralized source. Furthermore, the distribution of computational loads across clients allows DFL to offer larger scalability, enabling the involvement of a larger number of clients and even the support of larger-scale models without overburdening the central server.

Federated Averaging (FedAvg) (McMahan et al., 2017) is a widely adopted FL approach that disperses knowledge by averaging the parameters of models. Despite its advantages, FedAvg faces a significant limitation: the lack of support for model heterogeneity. This limitation becomes impractical in real-world scenarios where devices inherently possess diverse architectures. In a DFL system with model heterogeneity, FedAvg confines parameter averaging to models with the same architectures, thereby hindering knowledge sharing among clients. This problem exacerbates with the presence of heterogeneity in data between clients, affecting the preservation of global knowledge (Ye et al., 2023). Figure 1 demonstrates the adverse effects of model and data heterogeneity using decentralized FedAvg. This entails a need for a novel framework that better supports model and data heterogeneity in DFL. In this paper, we quantify global knowledge using global accuracy which is measured based on a global test dataset.

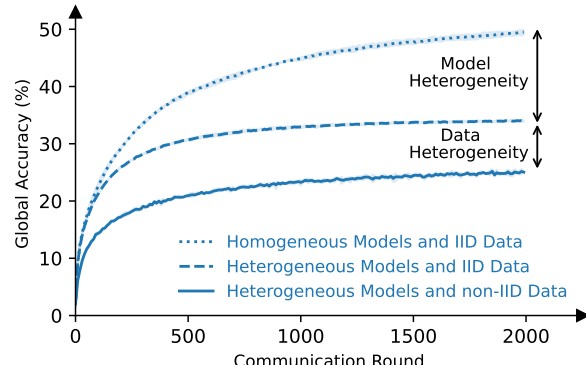

Figure 1: Demonstrating the adverse impact of model and data heterogeneity on global accuracy using decentralized FedAvg. The experiment uses CIFAR-100 dataset with 50 clients. Homogeneous models and IID data signify clients with identical model architectures and data distributions. In contrast, heterogeneous models and non-IID data indicate variations in both model architectures and data distributions among clients. Additional experimental details can be found in Section 4.4.1.

Researchers have extended FedAvg to support model heterogeneity, but these extensions often impose constraints on model architectures (Diao et al., 2020; Alam et al., 2022; Shen et al., 2020). Another approach that addresses model heterogeneity in FL involves mutual learning, where models collaboratively learn by teaching each other a task (Zhang et al., 2018; Li et al., 2021). Mutual learning enables knowledge transfer as models mimic each other's class probabilities. In this process, each client acts as both a student and a teacher. The objective function comprises a supervision loss component and another responsible for distilling knowledge from experts (Hinton et al., 2015). However, existing works utilizing knowledge distillation require a server or additional data (Lin et al., 2020; Li & Wang, 2019; Li et al., 2020a). Reliance on public data can be impractical, especially in sensitive domains such as health. Other works rely on generating synthetic data to perform knowledge transfer (Zhang et al., 2022b;a; Li et al., 2020b; Heinbaugh et al., 2023; Dai et al., 2024), which introduces an extra privacy concern. Thus, a solution that avoids using additional data is more desirable.

To this end, we propose a **D**ecentralized **F**ederated **M**utual **L**earning (DFML) framework that is 1) serverless, 2) supports model heterogeneity without imposing any architectural constraints, and 3) does not require additional public data. We define decentralized or serverless systems as lacking a dedicated, centralized client responsible for managing knowledge transfer. Table 1 highlights the advantages of our proposed DFML over prior arts. As will be shown in the results sec-

Table 1: Comparison between DFML and other FL methods.

| Framework | No Server | Nonrestrictive Heterogeneous | No Additional Data |
|---|---|---|---|
| FedAvg | ✗ | ✗ | ✓ |
| HeteroFL | ✗ | ✗ | ✓ |
| FedRolex | ✗ | ✗ | ✓ |
| FML | ✗ | ✗ | ✓ |
| FedDF | ✗ | ✓ | ✗ |
| FedMD | ✗ | ✓ | ✗ |
| Def-KT | ✓ | ✗ | ✓ |
| **DFML (Ours)** | ✓ | ✓ | ✓ |

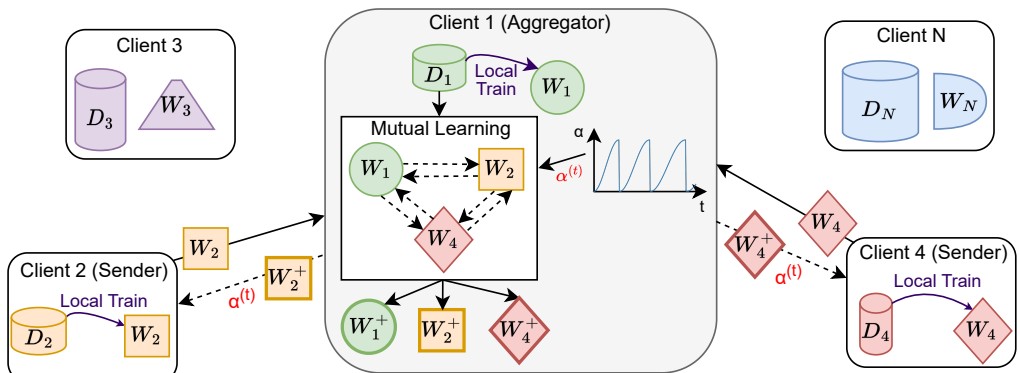

Figure 2: Our proposed DFML framework. In each communication round $t$, randomly selected clients (senders) send their locally trained models $W_n$ to another randomly chosen client (aggregator). Mutual learning takes place at the aggregator using $\alpha^{(t)}$. The updated models $W_n^+$ and $\alpha^{(t)}$ are then transmitted back to the senders. $\alpha^{(t)}$ controls the impact of the loss components in the objective function (see Section 3), and is computed based on a scheduler function. $t$ denotes the current communication round. Different shapes and sizes signify model and data heterogeneity. In this example, clients 2 and 4 act as senders, while client 1 serves as the aggregator.

tion, DFML outperforms other baselines in addressing the model and data heterogeneity problems in DFL. Figure 2 depicts our proposed DFML framework. In each communication round, multiple clients (senders) transmit their models to another client (aggregator) for mutual learning, effectively handling model heterogeneity. DFML leverages the aggregator's data for knowledge distillation, eliminating the need for extra data. Additionally, DFML addresses data heterogeneity by employing re-Weighted SoftMax (WSM) cross-entropy (Legate et al., 2023), which prevents models from drifting toward local objectives.

Furthermore, we observed varying performance levels[1] when using different fixed values to balance the ratio between the supervision and distillation loss components in the objective function. In response, we propose a cyclic variation (Loshchilov & Hutter, 2016; Smith, 2017) of the ratio between these loss components. The cyclical approach iteratively directs the model's objective between the aggregator's data and the reliance on experts to distill knowledge using that data. Gradually, as the distillation signal becomes dominant, global knowledge increases and eventually reaches a peak. This cyclical knowledge distillation further enhances global knowledge compared to using a fixed ratio between the supervision and distillation loss components.

In this paper, we empirically demonstrate the superior performance of DFML over state-of-the-art baselines in terms of convergence speed and global accuracy. The main contributions of this paper are summarized as follows:

- We propose a novel mutual learning framework that operates in DFL, supports nonrestrictive heterogeneous models and does not rely on additional data.

- We propose cyclically varying the ratio between the supervision and distillation signals in the objective function to enhance global accuracy.

## 2   Related Work

Due to the vastness of the FL literature, we restrict our review to works most relevant to our research. This includes studies that support homogeneous and heterogeneous architectures, operate in DFL, address catastrophic forgetting, and adapt knowledge distillation.

---

[1]Refer to Section 5.2

## 2.1 Homogeneous and Restrictive Heterogeneous Support

FL aims to learn global knowledge from distributed clients without sharing their private data. FedAvg (McMahan et al., 2017) uses a server to perform parameter averaging on locally updated models, but its support is limited to homogeneous architectures due to the nature of parameter averaging. In contrast, decentralized FedAvg (Li et al., 2021; Roy et al., 2019; Giuseppi et al., 2022; Savazzi et al., 2020) depends on client-to-client communication, with aggregation occurring on any participating clients. While other methods support heterogeneous models through parameter averaging, however, their support is constrained. Methods like Federated Dropout (Caldas et al., 2018), HeteroFL (Diao et al., 2020), and FedRolex (Alam et al., 2022) employ partial training with a random, static, and rolling technique for sub-model extractions, respectively. Lastly, FML (Shen et al., 2020) conducts mutual learning between personalized (heterogeneous) models, and a global (homogeneous) model. However, FML assumes the existence of a global model, and all the global knowledge resides in that model instead of the clients' heterogeneous models. This differs from our goal of transferring global knowledge to each of the clients' heterogeneous models.

## 2.2 Nonrestrictive Heterogeneous Support

Works that support model heterogeneity without imposing constraints exist, however, they require assistance from a server and the availability of public data (Li & Wang, 2019; Lin et al., 2020). FedMD (Li & Wang, 2019) assumes the availability of public data on all clients. In FedMD, clients generate predictions using public data which are then communicated to the server for averaging. Then, the averaged predictions are communicated back to the clients and are used to update the heterogeneous models using knowledge distillation. FedDF (Lin et al., 2020) communicates heterogeneous models to a server, where prototypes are assumed to exist. The server facilitates parameter averaging of models with same architectures, and knowledge distillation using unlabelled public data. The reliance on a server and additional data limits the applicability of these methods.

## 2.3 Decentralized FL

Def-KT (Li et al., 2021) operates within a DFL framework, where clients communicate models to other clients (aggregators) for mutual learning. Despite its serverless nature, Def-KT only supports homogeneous architectures as it replaces the aggregator's model with the incoming model. This hinders its effectiveness in scenarios with model heterogeneity.

## 2.4 Catastrophic Forgetting

Catastrophic forgetting focuses on acquiring new tasks without forgetting previous knowledge (Wang et al., 2023; De Lange et al., 2021). In the context of FL among clients with non-IID data distributions, catastrophic forgetting occurs, preventing models from reaching optimal global accuracy. To mitigate this issue, researchers aim to shield the learned representations from drastic adaptation. Asymmetric cross-entropy (ACE) is applied to the supervision signal of the objective function to address the representation drift problem (Caccia et al., 2021). ACE uses masked softmax on the classes that do not exist in the current data. Another approach, proposed by Legate et al. (2023), modifies the loss function of each client using re-Weighted SoftMax (WSM) cross-entropy, with re-weighting based on each client's class distribution.

## 2.5 Adaptive Knowledge Distillation

In the literature, existing works have explored scaling the loss function in knowledge distillation (Zhou et al., 2021; Clark et al., 2019); however, the advancements in this area have been minimal. WSL (Zhou et al., 2021) handles sample-wise bias-variance trade-off during distillation, while ANL-KD (Clark et al., 2019) gradually transitions the model from distillation to supervised learning. Early in training, the model primarily distills knowledge to leverage a useful training signal, and towards the end of the training, it relies more on labels to surpass its teachers. Additionally, although FedYogi (Reddi et al., 2020) proposes adaptive optimization, but its reliance on a server limits its applicability.

## 3 Proposed Approach

To begin with, we outline the DFL system within which our proposed DFML operates. Then, we provide a detailed explanation of DFML without delving into the discussion of varying the balance between the loss components. Lastly, we explain cyclical knowledge distillation and peak models.

### 3.1 System Setup

In our DFL setup, there are $N$ clients, each client $n$ is equipped with local training data $D_n \in \{D_1, D_2, ..., D_N\}$, regular model $W_n \in \{W_1, W_2, ..., W_N\}$, peak model $\widehat{W}_n \in \{\widehat{W}_1, \widehat{W}_2, ..., \widehat{W}_N\}$, and local weight alpha $\alpha_n \in \{\alpha_1, \alpha_2, ..., \alpha_N\}$. In each communication round, a client is chosen at random to act as the aggregator $a$. The choice of the aggregator is determined by the previous aggregator from the preceding round. In the initial round, the client with the lowest ID number is assigned the role of the aggregator. Additionally, several clients are randomly chosen as senders $S$ to send their models to the aggregator.

### 3.2 DFML Formulation

---
**Algorithm 1** DFML
---
**Input:** Initialize $N$ clients, each client $n$ has data $D_n$ and two models: regular $W_n$ and peak $\widehat{W}_n$. The local weight alpha of each client $\alpha_n = 0$.
**for** communication round $t = 1, 2, ..., T$ **do**
    Randomly select one aggregator $a \in \{1, ..., N\}$
    Randomly select senders $S \subset \{1, ..., N\}, a \notin S$
    Participants $\mathcal{P} = S \cup \{a\}$
    \\ Client Side
    **for all** $n \in \mathcal{P}$ **do**
        **for all** batch $X_n \in$ local data $D_n$ **do**
            $W_n \leftarrow$ locally train $W_n$ using Equation 2
    Send locally updated models $W_s$ for all $s \in S$ to $a$
    \\ Aggregator Side
    $\alpha^{(t)} \leftarrow$ scheduler$(\cdot)$
    **for** $k = 1, 2, ..., K$ **do**
        **for all** batch $X_a \in$ local data $D_a$ **do**
            **for all** $n \in \mathcal{P}$ **do**
                $z_n \leftarrow$ logits$(W_n, X_a)$
            **for all** $n \in \mathcal{P}$ **do**
                Update $W_n$ using $\alpha^{(t)}$ and logits according to Equation 1
    Send back updated models $W_s^+$ and $\alpha^{(t)}$ for all $s \in S$
    \\ Client Side
    **for all** $n \in \mathcal{P}$ **do**
        $W_n \leftarrow W_n^+$   \\ Update regular model
        **if** $\alpha^{(t)} \geq \alpha_n$ **then**
            $\widehat{W}_n \leftarrow W_n^+$   \\ Update peak model
            Set local weight $\alpha_n \leftarrow \alpha^{(t)}$
---

The goal of DFML is to enable the exchange of local information among clients without the need to share raw data. DFML facilitates knowledge sharing across multiple communication rounds $T$ until convergence. In each round $t \in T$, a set of participants ($\mathcal{P} = S \cup \{a\}$) is randomly selected. For each selected clients $n \in \mathcal{P}$, their regular models $W_n$ undergo local training on their private data $D_n$. This ensures that $W_n$ retains its local knowledge, allowing knowledge transfer to other participants during the aggregation process. The process of distilling knowledge to other models is referred to as aggregation. Subsequently, all locally trained models are sent to the aggregator for the aggregation process. When multiple clients send their models to the aggregator, multiple experts contribute during aggregation, enhancing the accuracy of the global knowledge transfer.

DFML employs weighted mutual learning for aggregation to allow models to collaboratively learn from each other using the aggregator's data. This technique ensures that larger models contribute more significantly to knowledge transfer compared to smaller models, leveraging their finer knowledge. DFML conducts the aggregation process $K$ times to maximize knowledge transfer without impacting the communication cost per round. Following this, all updated models $W_n^+$ are transmitted back to their respective senders. Subsequently, each participant $n \in \mathcal{P}$ replaces its model $W_n$ with the updated version $W_n^+$. This entire process is repeated until global knowledge has been disseminated across the entire network. Cyclical knowledge distillation and the peak models are explained in Section 3.2.1. Algorithm 1 describes our proposed DFML framework. The objective function of DFML is as follows:

$$\mathcal{L} = (1 - \alpha)\mathcal{L}_{\text{WSM}} + \alpha\mathcal{L}_{\text{KL}}, \tag{1}$$

where $\mathcal{L}_{\text{WSM}}$ represents the supervision loss signal computed using re-weighted softmax cross-entropy (WSM) (Legate et al., 2023), and $\mathcal{L}_{\text{KL}}$ represents the distillation loss signal computed by Kullback–Leibler Divergence (KL). The hyperparameter $\alpha$ controls the balance between these two loss components. $\mathcal{L}_{\text{WSM}}$ is defined as follows:

$$\mathcal{L}_{\text{WSM}} = -\sum_{x \in X_a}\left[z_n(x)_{y(x)} - \log\left(\sum_{c \in C}\beta_c e^{z_n(x)_c}\right)\right], \tag{2}$$

where $X_a$ represents a data batch drawn from the distribution $D_a$ of aggregator $a$. $z_n$ denotes the logits of data sample $x$ with model weights $W_n$ of client $n$, $y(x)$ is the data label, $\beta_c$ is a vector representing the proportion of label $c$ present in the aggregator's dataset, and $C$ is the set of classes in the entire dataset. During the aggregation process, $\mathcal{L}_{\text{WSM}}$ has access to only the aggregator's data and is computed for each $W_n$ available at the aggregator. However, during the local training stage, i.e. before the models are sent to the aggregator, $\mathcal{L}_{\text{WSM}}$ is also used at each client $n$, exploiting its private data to undergo local training.

The distillation loss component $\mathcal{L}_{\text{KL}}$ is defined as follows:

$$\mathcal{L}_{\text{KL}} = \sum_{x \in X_a}\sum_{q \neq n}^{\mathcal{P}}\left[\frac{\Phi_q}{\sum_{u \neq n}^{\mathcal{P}}\Phi_u}\text{KL}\big(p_q(x) \,||\, p_n(x)\big)\right], \tag{3}$$

where $X_a$ corresponds to a data batch drawn from the distribution $D_a$ of aggregator $a$. $\mathcal{P}$ denotes the set of participants in mutual learning, including the senders and the aggregator. $\Phi$ represents the model size based on the number of trainable parameters. Finally, $p_q(x)$ and $p_n(x)$ are the teacher $q$ and student $n$ predictions of the data sample $x$ with model weights $W_q$ and $W_n$, respectively. $u$ is a dummy variable indexing all teachers.

The use of WSM in both local training and mutual learning serves as a protective measure against catastrophic forgetting, which arises from shifts in data distribution between clients. WSM ensures that the models update parameters considering the proportion of available labels. This strategy prevents models from altering their accumulated knowledge on labels that are not present in the current data distribution, thereby safeguarding against catastrophic forgetting.

### 3.2.1 Cyclic knowledge distillation

Cyclical knowledge distillation is manifested by periodically adjusting the value of $\alpha$ in the objective function during each communication round. Inspired by (Loshchilov & Hutter, 2016; Smith, 2017; Izmailov et al., 2018), we use the cyclical behavior to vary $\alpha$. Cosine annealing scheduler, defined in Equation 4, is used to generate the cyclical behavior. This dynamic variation in $\alpha$ occurs with each new aggregator selection, leading to mutual learning with a distinct $\alpha$ value at each round. The cyclical process, influencing the balance between the supervision and distillation loss components, contributes to an overall increase in global knowledge. The global knowledge is measured by global accuracy throughout training.

$$\alpha^{(t)} = \alpha_{min} + \frac{1}{2}(\alpha_{max} - \alpha_{min})(1 + \cos(\frac{t}{T}\pi)), \tag{4}$$

where $\alpha^{(t)}$ is the $\alpha$ value at the current communication round $t$, $T$ is the maximum communication round, while $\alpha_{min}$ and $\alpha_{max}$ are the ranges of the $\alpha$ values.

Figure 3 depicts the impact of cyclical $\alpha$ on global accuracy. When the supervision signal dominates, each $W_n$ exclusively learns from $D_a$ without collaborating with other models (experts). This focus on the supervision signal directs the model's objective toward $D_a$, causing $W_n$ to lose previously acquired global knowledge from earlier rounds. As the distillation signal gains prominence, $W_n$ begins to reacquire global knowledge, facilitating simultaneous knowledge distillation among all models. With a dominant distillation signal, each model exclusively learns from other experts, maximizing global knowledge in each one.

Therefore, the peak in global accuracy is reached when the distillation signal is dominant ($\alpha$ is maximum), and the lowest is attained when the supervision signal is dominant ($\alpha$ is minimum). We observed that cyclically changing between the two signals leads to a higher global accuracy compared to using either one exclusively or a linear combination of them. This cyclical adjustment of $\alpha$ is crucial for the continuous growth in global accuracy throughout training, albeit with fluctuations in global accuracy.

$$\mathcal{L} = (1 - \alpha)\big(\text{Supervision signal}\big) + \alpha\big(\text{Distillation signal}\big)$$

Figure 3: Illustrating the impact of cyclically varying $\alpha$ on global accuracy. Peak models are updated up to the first $\alpha$ maximum and every subsequent time $\alpha$ reaches its maximum limit. In this example, $\alpha$ is varied using a cosine annealing scheduler.

### 3.2.2 Peak models

To counteract the undesirable fluctuations in global accuracy resulting from the cyclical process, we introduce an additional model for each client, termed the *peak model* $\widehat{W}_n$. The primary role of $\widehat{W}_n$ is to retain the best global parameters of $W_n$. Specifically, each $\widehat{W}_n$ is updated whenever $W_n$ is aggregated with a dominant distillation signal. $\widehat{W}_n$ are detached from the training process and are kept in a frozen state, preserving the maximum global accuracy achieved so far. Also, the peak models are continuously updated from the initial communication round up to the first $\alpha$ maximum, allowing them to quickly reach the first global accuracy peak. Thus, the peak models act as a stabilizing mechanism, retaining the optimal global knowledge attained.

## 4 Experiments

### 4.1 Dataset

We evaluate our proposed DFML against prevalent baselines using five datasets including CIFAR-10/100, FMNIST, Caltech101, Oxford Pets, and Stanford Cars. The evaluation covers experiments on two data distribution shifts: Independent and Identically Distributed (IID) and non-IID. The non-IID distribution involves a heavy label shift based on the Dirichlet distribution with $\beta = 0.1$. The dataset is distributed among clients by dividing the train set into $N$ splits, either evenly for IID or utilizing Dirichlet distribution for non-IID. Each split is further segmented into training and validation sets following an 80:20 ratio. For Caltech101, samples are first split 80:20, where the 20% represents the global test set, and the remaining samples follows the defined splitting strategy above. Validation sets are employed to assess local performance, while the entire test set evaluates the global accuracy of the clients. The global accuracy of DFML is evaluated by examining the peak models unless stated otherwise. The data partitions for all clients are available in Appendix A.1.

### 4.2 Implementations

In our experiments, we utilize CNN, ResNet, ViT, and EfficientNet as our model architectures. The evaluation of DFML encompasses three architecture modes: homogeneous, restrictive heterogeneous, and nonrestrictive heterogeneous. We name these three modes: $H0$, $H1$, and $H2$, respectively. Details of these modes and the associated model architectures are outlined in Table 10. To ensure a fair comparison, all experiments including the baselines are run with WSM instead of Cross-Entropy (CE), unless specified otherwise. Further implementation details are available in Appendix A.2.

Table 2: Different model specifications supported by the baselines and DFML in the network.

| Framework | Different Model Types | Different # of Layers | Different Width of layers |
|---|---|---|---|
| Dec. FedAvg | ✗ | ✗ | ✗ |
| Dec. FedProx | ✗ | ✗ | ✗ |
| Def-KT | ✗ | ✗ | ✗ |
| Dec. HeteroFL | ✗ | ✗ | ✓ |
| Dec. FedRolex | ✗ | ✗ | ✓ |
| **DFML (Ours)** | ✓ | ✓ | ✓ |

## 4.3 Baselines

To address the absence of baselines tailored for DFL settings, we derive baselines by decentralizing some state-of-the-art CFL algorithms, adapting them to function within our DFL system. We begin our experiments by constraining model architectures to enable comparison with existing baselines. The initial experiments focus on homogeneous architectures for direct comparison with baselines like decentralized FedAvg, FedProx (Li et al., 2020c) and Def-KT. Our derived decentralized version of FedAvg and FedProx are referred to as decentralized FedAvg and FedProx, respectively. Def-KT intrinsically operates in a DFL framework.

Subsequently, we conduct experiments with restrictive heterogeneous architectures, where models have the same number of layers but different widths. This allows comparison of DFML with our derived decentralized versions of partial training methods (HeteroFL and FedRolex) alongside FedAvg. Further details on the derived decentralized baselines are provided in Appendix A.3.

Following this, we demonstrate the full capabilities of DFML by conducting experiments with nonrestrictive heterogeneous architectures, which are incompatible with partial training algorithms. The only baseline available for this set of experiments is decentralized FedAvg. In this paper, we omit the comparison of DFML with baselines that require additional data such as (Lin et al., 2020; Li & Wang, 2019) to ensure fairness.

Table 2 summarizes the model features supported by the baselines and our proposed DFML in the network. Decentralized FedAvg, decentralized FedProx and Def-KT baselines only accommodate homogeneous models, requiring all models to be of same type, with the same number of layers, and each layer have the same dimensions. Decentralized HeteroFL and FedRolex baselines support models of the same type, with the same number of layers, but they allow layers to have different widths. In contrast, our proposed DFML can support different model types, models with different number of layers, and varying widths.

## 4.4 Results

We evaluate DFML against other state-of-the-art baselines. We first demonstrate the effectiveness of DFML in handling model and data heterogeneity. Second, we prove that DFML outperforms all baselines in terms of final convergence speed and final global accuracy across three architecture modes: homogeneous, restrictive heterogeneous, and nonrestrictive heterogeneous architectures. Next, we demonstrate the performance of each cluster of architectures in an experiment with nonrestrictive heterogeneous architectures. Finally, we present the scalability of DFML under a significant model heterogeneity scenario.

### 4.4.1 Model and Data Heterogeneity

Figure 4 demonstrates that our proposed DFML under model and data heterogeneity, mitigates the impact on global accuracy more effectively than decentralized FedAvg and HeteroFL. To ensure a fair comparison between homogeneous and heterogeneous experiments, we maintained the same average number of parameters in each case. This was achieved by selecting the median model of the five different architectures from the heterogeneous experiment as the model used in the homogeneous experiment.

### 4.4.2 Homogeneous Architectures

Table 3 demonstrates that our DFML outperforms decentralized FedAvg, decentralized FedProx, and Def-KT in terms of global accuracy across datasets and under two data distributions. Larger improvements are

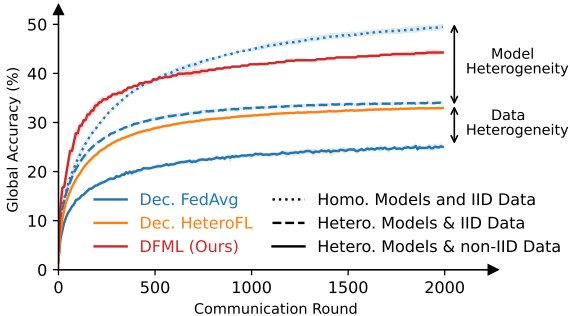

Figure 4: Demonstrating the global accuracy gain DFML achieves in comparison with decentralized FedAvg and HeteroFL under model and data heterogeneity. CIFAR-100 dataset is used with 50 clients and CNN architectures.

Table 3: Global accuracy comparison using homogeneous CNN architectures with 50 clients and 25 senders. For Def-KT, 25 aggregators are selected, and 1 aggregator for the other methods.

| Method | CIFAR-10 | | CIFAR-100 | |
|---|---|---|---|---|
| | IID | non-IID | IID | non-IID |
| Dec. FedAvg | $80.00_{\pm 0.19}$ | $74.68_{\pm 0.06}$ | $48.32_{\pm 0.18}$ | $43.17_{\pm 0.16}$ |
| Dec. FedProx | $79.93_{\pm 0.04}$ | $74.55_{\pm 0.19}$ | $47.97_{\pm 0.34}$ | $42.41_{\pm 0.47}$ |
| Def-KT | $79.29_{\pm 0.18}$ | $72.59_{\pm 0.34}$ | $48.43_{\pm 0.18}$ | $43.84_{\pm 0.24}$ |
| **DFML (Ours)** | $\mathbf{80.96}_{\pm 0.07}$ | $\mathbf{76.27}_{\pm 0.40}$ | $\mathbf{50.47}_{\pm 0.29}$ | $\mathbf{46.41}_{\pm 0.48}$ |

Table 4: Global accuracy comparison using restrictive heterogeneous architectures with 50 clients and 25 senders. Architectures used are CNN, ResNet, and ViT.

| Method | CNN | | | | ResNet | | | | ViT | | | |
|---|---|---|---|---|---|---|---|---|---|---|---|---|
| | CIFAR-10 | | CIFAR-100 | | CIFAR-10 | | CIFAR-100 | | CIFAR-10 | | CIFAR-100 | |
| | IID | non-IID | IID | non-IID | IID | non-IID | IID | non-IID | IID | non-IID | IID | non-IID |
| Dec. FedAvg | $72.09_{\pm 0.20}$ | $59.84_{\pm 0.18}$ | $33.59_{\pm 0.06}$ | $27.03_{\pm 0.23}$ | $79.84_{\pm 0.16}$ | $54.93_{\pm 0.76}$ | $39.98_{\pm 0.45}$ | $28.80_{\pm 0.24}$ | $64.50_{\pm 0.13}$ | $50.76_{\pm 0.26}$ | $27.43_{\pm 0.22}$ | $22.27_{\pm 0.03}$ |
| Dec. HeteroFL | $73.96_{\pm 0.21}$ | $64.21_{\pm 0.67}$ | $37.62_{\pm 0.16}$ | $32.98_{\pm 0.10}$ | $81.70_{\pm 0.11}$ | $68.92_{\pm 0.60}$ | $43.33_{\pm 0.34}$ | $34.92_{\pm 0.29}$ | $65.08_{\pm 0.61}$ | $53.90_{\pm 0.09}$ | $29.60_{\pm 0.53}$ | $25.65_{\pm 0.25}$ |
| Dec. FedRolex | $74.38_{\pm 0.02}$ | $64.30_{\pm 0.41}$ | $37.65_{\pm 0.27}$ | $32.83_{\pm 0.16}$ | $81.70_{\pm 0.17}$ | $69.41_{\pm 0.19}$ | $43.30_{\pm 0.20}$ | $34.94_{\pm 0.43}$ | $65.17_{\pm 0.40}$ | $53.56_{\pm 0.61}$ | $30.44_{\pm 0.21}$ | $25.81_{\pm 0.06}$ |
| **DFML (Ours)** | $\mathbf{79.02}_{\pm 0.04}$ | $\mathbf{73.87}_{\pm 0.19}$ | $\mathbf{48.60}_{\pm 0.03}$ | $\mathbf{44.26}_{\pm 0.20}$ | $\mathbf{85.68}_{\pm 0.01}$ | $\mathbf{71.24}_{\pm 0.02}$ | $\mathbf{53.32}_{\pm 0.07}$ | $\mathbf{46.27}_{\pm 0.38}$ | $\mathbf{68.38}_{\pm 0.03}$ | $\mathbf{54.50}_{\pm 0.01}$ | $\mathbf{36.48}_{\pm 0.20}$ | $\mathbf{28.56}_{\pm 0.60}$ |

Table 5: Global accuracy comparison using restrictive heterogeneous architectures. The experiments are conducted using ResNet architectures with 50 clients and 25 senders.

| Method | FMNIST | | Caltech101 | | Oxford Pets | | Stanford Cars | |
|---|---|---|---|---|---|---|---|---|
| | IID | non-IID | IID | non-IID | IID | non-IID | IID | non-IID |
| Dec. FedAvg | $89.07_{\pm 0.05}$ | $73.53_{\pm 0.13}$ | $36.74_{\pm 0.61}$ | $22.83_{\pm 0.34}$ | $10.25_{\pm 0.43}$ | $9.21_{\pm 0.31}$ | $2.83_{\pm 0.24}$ | $3.44_{\pm 0.21}$ |
| Dec. HeteroFL | $91.06_{\pm 0.15}$ | $85.49_{\pm 0.10}$ | $49.05_{\pm 0.21}$ | $38.04_{\pm 0.16}$ | $23.35_{\pm 0.14}$ | $17.68_{\pm 0.12}$ | $5.55_{\pm 0.14}$ | $6.25_{\pm 0.10}$ |
| Dec. FedRolex | $91.26_{\pm 0.20}$ | $86.06_{\pm 0.19}$ | $49.14_{\pm 0.32}$ | $37.75_{\pm 0.22}$ | $22.67_{\pm 0.14}$ | $18.39_{\pm 0.15}$ | $5.88_{\pm 0.11}$ | $6.09_{\pm 0.09}$ |
| **DFML (Ours)** | $\mathbf{92.01}_{\pm 0.02}$ | $\mathbf{87.31}_{\pm 0.03}$ | $\mathbf{60.05}_{\pm 0.04}$ | $\mathbf{50.07}_{\pm 0.06}$ | $\mathbf{40.42}_{\pm 0.16}$ | $\mathbf{31.25}_{\pm 0.11}$ | $\mathbf{41.12}_{\pm 0.10}$ | $\mathbf{25.31}_{\pm 0.09}$ |

recorded under non-IID data shifts. To align the number of communications per round for all baselines, adjustments were made for Def-KT by setting the number of aggregators to 25. This is necessary as in Def-KT each sending model should be received by a different aggregator. Decentralized FedProx is not explored further in the remaining experiments as it yielded approximately the same performance as decentralized FedAvg under homogeneous architectures. Centralized FedProx requires the availability of a global model at each participating client to restrict local updates to stay close to the global model. However, in a decentralized setting, a single global model does not exist, and each client treats its local model, before local training, as a global model. This limitation explains why FedProx did not outperform FedAvg in the decentralized setting.

### 4.4.3 Heterogeneous Architectures with Restrictions

In this set of experiments, some restrictions are applied to heterogeneous architectures. Tables 4 and 5 demonstrate that DFML consistently outperforms all baselines in terms of global accuracy across different architectures, datasets, and both IID and non-IID data distributions. Another observation is that the partial training baselines outperformed decentralized FedAvg, which is expected as more knowledge is shared by averaging overlapping parameters of the models rather than only averaging models with the same architec-

Table 6: Global accuracy comparison using nonrestrictive heterogeneous architectures. The experiments are conducted using CNN architectures. Different numbers of clients $N$ are used, with $50\% \times N$ as senders.

| | $N:10$ | | | | $N:50$ | | | | $N:100$ | | | |
|---|---|---|---|---|---|---|---|---|---|---|---|---|
| | CIFAR-10 | | CIFAR-100 | | CIFAR-10 | | CIFAR-100 | | CIFAR-10 | | CIFAR-100 | |
| Method | IID | non-IID | IID | non-IID | IID | non-IID | IID | non-IID | IID | non-IID | IID | non-IID |
| Dec. FedAvg | $72.65_{\pm 0.17}$ | $43.78_{\pm 0.06}$ | $33.58_{\pm 0.08}$ | $23.01_{\pm 0.09}$ | $71.51_{\pm 0.24}$ | $59.05_{\pm 0.22}$ | $33.19_{\pm 0.19}$ | $26.27_{\pm 0.30}$ | $71.00_{\pm 0.32}$ | $57.79_{\pm 0.11}$ | $32.30_{\pm 0.30}$ | $26.93_{\pm 0.13}$ |
| DFML (Ours) | $\mathbf{83.87}_{\pm 0.22}$ | $\mathbf{74.30}_{\pm 0.88}$ | $\mathbf{54.03}_{\pm 0.15}$ | $\mathbf{49.67}_{\pm 0.20}$ | $\mathbf{81.74}_{\pm 0.04}$ | $\mathbf{75.51}_{\pm 0.12}$ | $\mathbf{50.39}_{\pm 0.09}$ | $\mathbf{46.22}_{\pm 0.07}$ | $\mathbf{79.94}_{\pm 0.03}$ | $\mathbf{71.75}_{\pm 0.05}$ | $\mathbf{47.66}_{\pm 0.06}$ | $\mathbf{42.84}_{\pm 0.23}$ |

Table 7: Global accuracy comparison using nonrestrictive heterogeneous architectures. The experiments are conducted using EfficientNet architectures with 10 clients and 5 senders.

| | Caltech101 | | Oxford Pets | | Stanford Cars | |
|---|---|---|---|---|---|---|
| Method | IID | non-IID | IID | non-IID | IID | non-IID |
| Dec. FedAvg | $54.75_{\pm 0.24}$ | $30.92_{\pm 0.16}$ | $18.34_{\pm 0.31}$ | $15.34_{\pm 0.21}$ | $6.07_{\pm 0.25}$ | $11.83_{\pm 0.12}$ |
| DFML (Ours) | $\mathbf{83.42}_{\pm 0.20}$ | $\mathbf{65.70}_{\pm 0.13}$ | $\mathbf{68.90}_{\pm 0.22}$ | $\mathbf{42.94}_{\pm 0.13}$ | $\mathbf{70.29}_{\pm 0.19}$ | $\mathbf{50.63}_{\pm 0.23}$ |

Table 8: Overall communication rounds DFML requires to attain the same accuracy as decentralized FedAvg achieves at communication rounds 100 and 500. Both methods have the same communication cost per round. Different numbers of clients $N$ are used.

| | | $N:10$ | | | | $N:50$ | | | | $N:100$ | | | |
|---|---|---|---|---|---|---|---|---|---|---|---|---|---|
| | Dec. FedAvg | CIFAR-10 | | CIFAR-100 | | CIFAR-10 | | CIFAR-100 | | CIFAR-10 | | CIFAR-100 | |
| Method | Communication Round | IID | non-IID | IID | non-IID | IID | non-IID | IID | non-IID | IID | non-IID | IID | non-IID |
| DFML (Ours) | 100 | 10 | 20 | 20 | 10 | 30 | 40 | 30 | 30 | 50 | 60 | 40 | 50 |
| | 500 | 20 | 20 | 20 | 20 | 90 | 90 | 90 | 80 | 150 | 190 | 130 | 130 |

Table 9: Global accuracy comparison between DFML and decentralized FedAvg with different supervision signals and cyclically varying $\alpha$. The dataset used is CIFAR-100, with 50 clients and 25 senders.

| | | | CNN | | ResNet | | ViT | |
|---|---|---|---|---|---|---|---|---|
| Method | CE / WSM | Cyclical $\alpha$ | IID | non-IID | IID | non-IID | IID | non-IID |
| DFML (Ours) | CE | ✗ | $47.44_{\pm 0.08}$ | $29.16_{\pm 2.27}$ | $49.13_{\pm 0.30}$ | $18.78_{\pm 0.67}$ | $35.66_{\pm 0.02}$ | $14.02_{\pm 0.01}$ |
| | CE | ✓ | $48.44_{\pm 0.23}$ | $41.47_{\pm 0.30}$ | $\mathbf{55.29}_{\pm 0.13}$ | $41.07_{\pm 1.27}$ | $\mathbf{36.62}_{\pm 0.30}$ | $22.34_{\pm 0.12}$ |
| | WSM | ✗ | $47.57_{\pm 0.26}$ | $43.59_{\pm 0.35}$ | $49.77_{\pm 0.24}$ | $30.91_{\pm 0.96}$ | $34.04_{\pm 0.21}$ | $25.04_{\pm 0.56}$ |
| | WSM | ✓ | $\mathbf{48.60}_{\pm 0.03}$ | $\mathbf{44.26}_{\pm 0.20}$ | $53.32_{\pm 0.07}$ | $\mathbf{46.27}_{\pm 0.38}$ | $36.48_{\pm 0.20}$ | $\mathbf{28.56}_{\pm 0.60}$ |

tures. Furthermore, we implemented decentralized Federated Dropout; however, the results are not reported as it did not work. The poor performance is attributed to local models being assigned random parameters from the global model generated at the aggregator.

### 4.4.4 Heterogeneous Architectures without Restrictions

After confirming that our proposed DFML competes with the prevalent baselines in DFL, we showcase the strength of DFML in knowledge transfer using nonrestrictive heterogeneous models. Additionally, we evaluate DFML under different $N$ clients. From Tables 6 and 7, it is evident that across various $N$ clients, datasets, and data distributions, DFML consistently achieves superior global accuracy compared to the baseline. Table 8 shows that DFML requires fewer communication rounds to reach the same accuracy as decentralized FedAvg does at specific rounds. The communication cost per round for all experiments is 50, involving the transmission of 25 models to and from the aggregator. Appendix A.4 provides further experimental analysis on DFML. A visual illustration of the convergence speedup achieved by DFML compared to the baselines is presented in Appendix A.4.4. Moreover, a comparison between DFML and a decentralized version of FML is provided in Appendix A.4.11.

### 4.4.5 Performance per Cluster of Architectures

Figure 5 presents the performance of each architecture cluster in DFML compared to decentralized FedAvg. In this experiment, five different CNN architectures are distributed among 50 clients. The figure demonstrates that the bigger the architecture size, the higher the attained global accuracy. Moreover, all clusters in DFML surpasses their corresponding clusters in decentralized FedAvg.

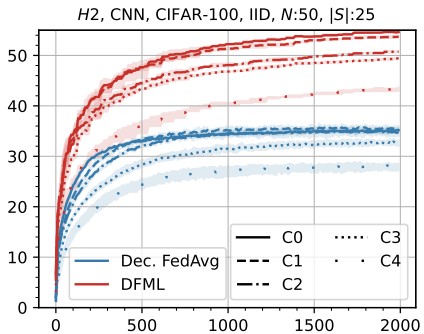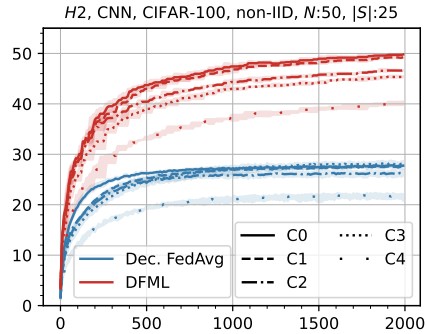

Figure 5: Performance comparison between the different architecture clusters in both DFML and decentralized FedAvg. In this experiment, five nonrestrictive heterogeneous architectures are distributed among 50 clients. C0, C1, C2, C3, and C4 represent the global accuracy average of all models with CNN architectures [32, 64, 128, 256], [32, 64, 128], [32, 64], [16, 32, 64], and [8, 16, 32, 64], respectively.

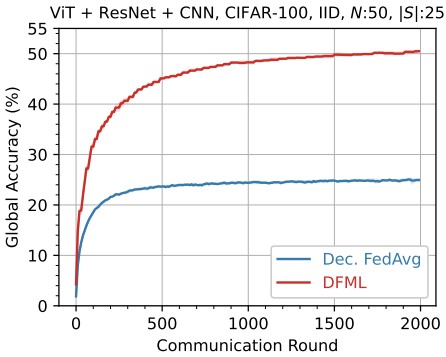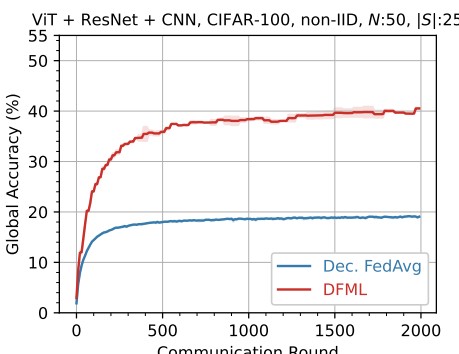

Figure 6: Comparison between DFML and decentralized FedAvg under significant model heterogeneity. Ten different architectures are distributed among 50 clients including: 2 ViT, 4 ResNet, and 4 CNN architectures.

### 4.4.6 High Model Heterogeneity

To showcase the scalability of DFML with model heterogeneity, we conduct an experiment involving 50 clients with significant model heterogeneity. We compare the results obtained by DFML to decentralized FedAvg. In this experiment, ten different architectures are deployed and selected from Table 10: the two largest ViT architectures, the four largest ResNet architectures, and the four largest CNN architectures under the $H2$ category. Figure 6 demonstrates that DFML performs effectively under heavy model heterogeneity conditions and greatly outperforms decentralized FedAvg.

## 5 Analysis

In this section, we first demonstrate the effect of using different fixed $\alpha$ values versus cyclically adjusting it. Second, we present the impact of cyclical $\alpha$ on global accuracy by evaluating the regular models. Subsequently, we examine the behavior exhibited by both regular and peak models as $\alpha$ changes, and highlight the ability of the peak models to capture the peaks in global accuracy. Last, we conduct an analysis to understand the effects of different supervision signals and cyclical $\alpha$ on global accuracy.

### 5.1 Regular vs Peak Models

Figure 7 shows the fluctuations in global accuracy as $\alpha$ is cyclically varied. When $\alpha = 0$, representing only the supervision signal in the objective function, global accuracy is at its lowest. This is attributed to models being optimized solely toward the aggregator's local data. However, as $\alpha$ increases and eventually reaches its maximum defined value, models gain knowledge from each other through knowledge distillation, resulting

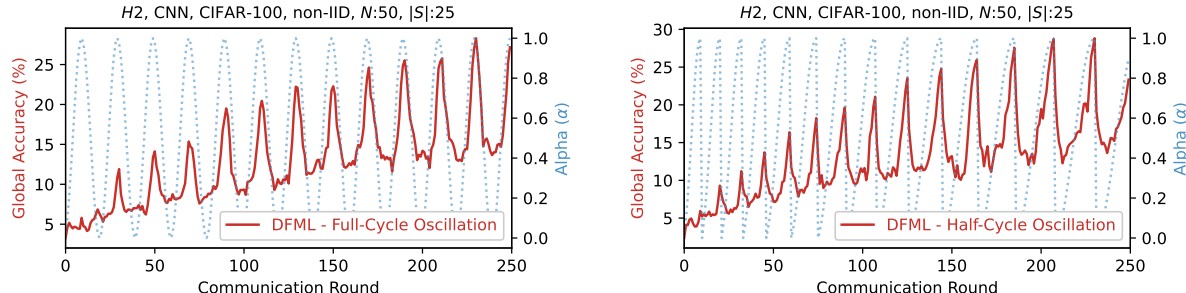

Figure 7: Performance comparison of the regular models against cyclically oscillating $\alpha$ with full and half cycles, respectively. The supervision signal used is CE.

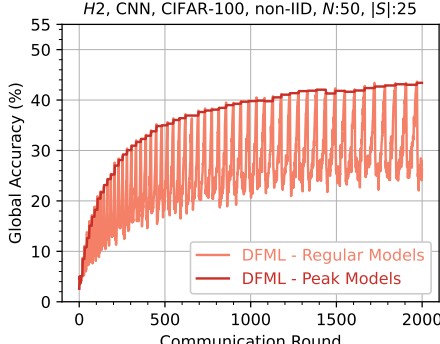

Figure 8: Performance comparison between DFML regular models, that are communicated and updated in each communication round, and peak models that are updated only when a peak occurs. The supervision signal used is CE.

in a peak in global accuracy. The decline in global accuracy starts when the distillation signal diminishes, and the supervisory signal takes over. To maintain global accuracy at the peaks, the peak models are used. Initially, the peak models are updated regularly with every regular model update until the first maximum $\alpha$ value is reached. After that, the peak models are only updated when the regular models are aggregated using the highest value of $\alpha$. Figure 8 depicts the global accuracy of both regular and peak models, highlighting the stability achieved by the peak models throughout training.

### 5.2 Fixed vs Cyclical $\alpha$

As shown in Figure 7, the highest global accuracy is consistently achieved when $\alpha$ reaches its maximum defined value. Now, we address the impact of using different fixed values for $\alpha$ and whether setting $\alpha = 1$ throughout training yields similar performance as cyclically changing $\alpha$. Figure 9 demonstrates that using different fixed values of $\alpha$, under different distribution shifts, results in varied performance levels. Furthermore, fixing $\alpha = 1$ leads to the worst global accuracy because when only the distillation signal is present throughout training without any supervision, noise will be propagated. This results in experts teaching each other incorrect information.

### 5.3 Supervision Signal and Cyclical $\alpha$

The impact of different supervision signals (CE and WSM) and cyclical $\alpha$ on DFML is presented in Table 9. Results indicate that the use of WSM primarily enhances the global accuracy in non-IID data distribution shifts. Furthermore, the addition of cyclical $\alpha$ on top of CE and WSM further improves global accuracy. The best outcomes are mostly reported when both WSM and cyclical $\alpha$ are applied. Additional experimental analysis on cyclical knowledge distillation can be found in Appendix A.5.

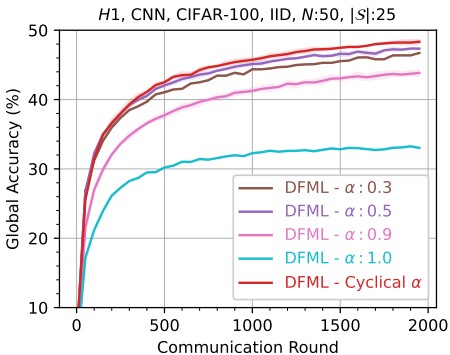 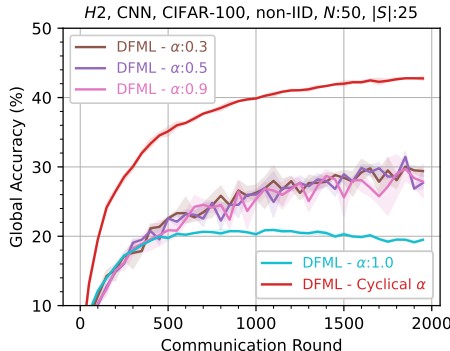

Figure 9: Performance comparison between different fixed $\alpha$ values and cyclically varying it, under IID and non-IID data distributions. The supervision signal used is CE.

## 6 Limitations

While DFML presents significant advancements over existing baselines, it does have certain limitations that need to be acknowledged. One notable limitation is its computational expense. DFML requires aggregators to possess sufficient memory to receive multiple models and enough computational power to perform mutual learning, involving both forward and backward passes on all models at the aggregator. Another limitation is the lack of convergence analysis within in this paper. While proving the convergence properties of DFML is beyond the current scope, proving the convergence properties of DFML is left for future work.

## 7 Conclusion

We proposed DFML, a framework that supports a decentralized knowledge transfer among heterogeneous models, without architectural constraints or reliance on additional data. DFML overcomes common centralization issues such as communication bottlenecks and single points of failure, making it a robust alternative for real-world applications. DFML outperformed state-of-the-art baselines in addressing model and data heterogeneity in DFL, showcasing better convergence speed and global accuracy.

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

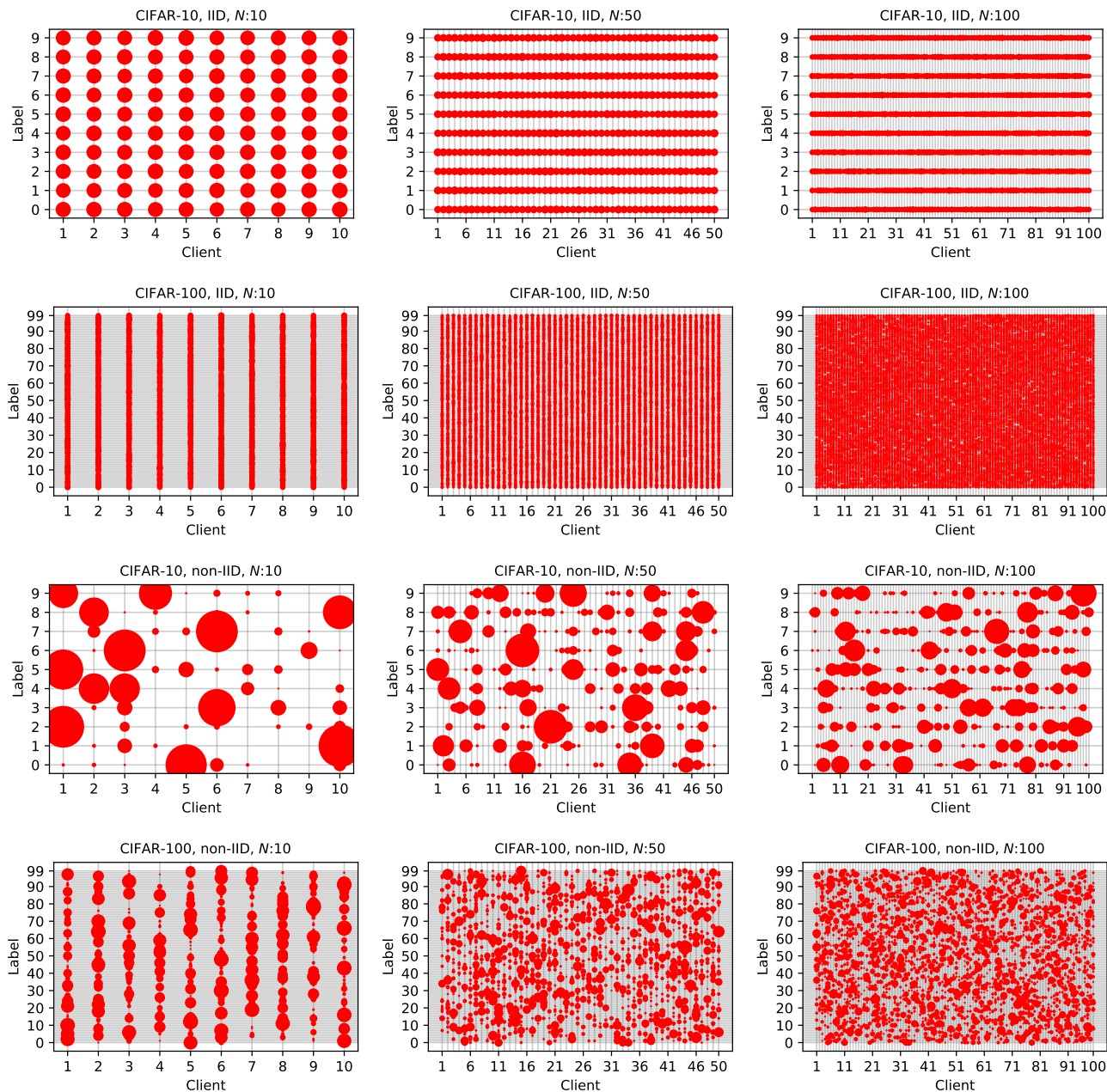

Figure 10: Data partitions based on IID and non-IID distribution with clients $N = \{10, 50, 100\}$ for both CIFAR-10 and CIFAR-100 datasets. The size of the red circle represents the magnitude of data samples for each class label in each client.

# A   Appendix

## A.1   Dataset Distributions

The data partitions for both IID and non-IID distributions of CIFAR-10 and CIFAR-100 datasets are illustrated in Figure 10.

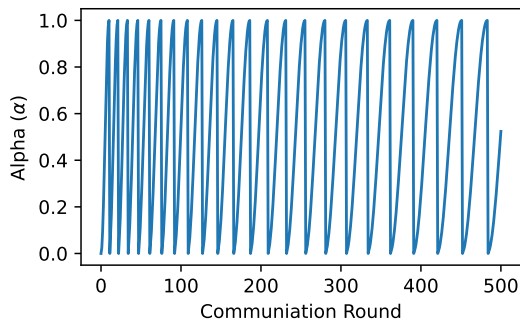

Figure 11: Cyclic oscillation of $\alpha$ with incremental period increase. $\alpha$ ranges from $0 \to 1$.

## A.2 Implementation Details

In our setup, we assume a star topology where all clients can send and receive models from any other client. Each communication round involves randomly selecting $50\%$ of the clients as senders $S$, with an additional client randomly chosen as the aggregator $a$ (unless specified otherwise). We utilize SGD optimizer for each client with momentum 0.9 and weight decay 5e-4. The learning rate is selected from $\{0.1, 0.01, 0.001\}$. The batch size is set to 8 for the EfficientNet experiments, 16 for the ResNet experiments using Caltech101, Oxford Pets, and StanfordCars datasets, and batch size of 64 is used for all other experiments,. For the cyclic $\alpha$ scheduler, we apply cosine annealing. The initial oscillating period is set to 10 and is incrementally increased after each completion. $\alpha$ is oscillated from 0 to a maximum value selected between $\{0.8, 0.9, 1.0\}$. Figure 11 illustrates an example of the behavior of $\alpha$ throughout training. The number of mutual learning epochs $K$, performed at the aggregator, is set to 10. Moreover, the temperature is configured to 1. All experiments are repeated for 3 trials with random seeds. With the existence of an $\alpha$ and period schedulers, aggregators need to be aware of communication round $t$, and the round when the period was last updated to compute the period and $\alpha$ value. To achieve this, each aggregator needs to communicate these two values to the next aggregator.

The architectures employed in our experiments are presented in Table 10. Modes $H0$, $H1$, and $H2$ refer to homogeneous, restrictive heterogeneous, and nonrestrictive heterogeneous architectures, respectively. In mode $H1$, the model rates for different model types are: $[2^0, 2^{-1}, 2^{-2}, 2^{-3}, 2^{-4}]$. Here, smaller models are scaled versions of the largest model in terms of width. Mode $H2$ designates heterogeneous architectures with no constraints, allowing each client to have a different number of layers and hidden channel sizes. In our $H1$ and $H2$ experiments, the models are evenly distributed among clients.

For CNNs, the values inside the array represent the number of channels in each layer, and the array's length corresponds to the number of layers. Each CNN layer has $5 \times 5$ kernels followed by the ReLU activation function, $2\times2$ max pooling, and layer normalization. In ResNets, the largest model is pre-activated ResNet18, while others are scaled versions of ResNet18. The values inside the array represent the number of channels per layer, and each layer consists of two blocks. Similarly, for ViTs, the largest model is comprised of 2 layers with 512 channels each; the others are scaled versions of the largest ViT. Shifted Patch Tokenization (SPT) and Locality Self-Attention (LSA) (Lee et al., 2021) are used in our ViT architectures to solve the lack of locality inductive bias and enable us to use non-pretrained ViTs on small datasets. Additionally, for ViTs, the patch size is set to $4 \times 4$, head dimensions to 64, depth to 2, dropout to 0.1, and embedding dropout to 0.1.

## A.3 Baselines

### A.3.1 Decentralized FedAvg

When senders send their heterogeneous models to the aggregator, decentralized FedAvg performs parameter averaging exclusively among models with identical architectures. In the homogeneous scenario, averaging encompasses all available models. In contrast, in heterogeneous scenarios, clusters of global models are

Table 10: Different architecture modes.

| Mode Name | Model | | |
|---|---|---|---|
| | Heterogeneity | Type | Architectures |
| $H0$ | Homogeneous | CNN | $[32, 64]$ |
| $H1$ | Restrictive Heterogeneous | CNN | $[128, 256]$ $[64, 128]$ $[32, 64]$ $[16, 32]$ $[8, 16]$ |
| | | ResNet | $[64, 128, 256, 512]$ $[32, 64, 128, 256]$ $[16, 32, 64, 128]$ $[8, 16, 32, 64]$ $[4, 8, 16, 32]$ |
| | | ViT | $[512, 512]$ $[256, 256]$ $[128, 128]$ $[64, 64]$ $[32, 32]$ |
| $H2$ | Nonrestrictive Heterogeneous | CNN | $[32, 64, 128, 256]$ $[32, 64, 128]$ $[32, 64]$ $[16, 32, 64]$ $[8, 16, 32, 64]$ |
| | | EfficientNet | B0 B1 B2 B3 B4 |

formed as identical models in each group are averaged together. The resulting global models are then communicated back to the clients with the same model architecture. During parameter averaging, weights are assigned based on the number of data samples in each client. Algorithm 2 provides a detailed explanation of decentralized FedAvg. Model parameters in FedAvg are aggregated as follows:

$$W_g = \frac{1}{\sum_{n \in \mathcal{P}} d_n} \sum_{n \in \mathcal{P}} d_n W_n, \tag{5}$$

where $W_g$ is the global model and $W_n$ is the model of client $n$. The weight $d_n$ is based on the number of data samples in the client.

### A.3.2 Decentralized FedProx

Decentralized FedProx is similar to decentralized FedAvg in that a subset of clients send their locally trained models to the aggregator for parameter averaging to form a global model, which is then sent back to the senders. However, FedProx adds a proximal term to the local training objective, which helps improve the method's stability by restricting the local updates to be closer to the initial (global) model. The proximal term necessitates keeping a copy of the global model while performing local training. In decentralized setting with partial client participation, multiple versions of global models exist throughout the network. Decentralizing FedProx requires each client to treat its model as the global model before performing local training. Algorithm 2 provides a detailed explanation of decentralized FedProx. The hyperparameter $\mu$ is selected from $\{0.5, 1, 2\}$.

---

**Algorithm 2** Decentralized FedAvg and FedProx

---

**Input:** Initialize $N$ clients, each client $n$ has a model $W_n$ and data $D_n$. All models with the same architectures have the same initialization.

**for** communication round $t = 1, 2, ..., T$ **do**

    Randomly select one aggregator $a \in \{1, ..., N\}$

    Randomly select senders $S \subset \{1, ..., N\}$, $a \notin S$

    Participants $\mathcal{P} = S \cup \{a\}$

    \\ Client Side

    **for all** $n \in \mathcal{P}$ **do**

        $\widetilde{W}_n \leftarrow W_n$

        **for all** batch $X_n \in$ local data $D_n$ **do**

            $W_n \leftarrow$ locally train $W_n$ using Equation 2

            $W_n \leftarrow$ locally train $W_n$ using Equation $2 + \frac{\mu}{2} \left\| \widetilde{W}_n - W_n \right\|^2$

    Send locally updated models $W_s$ for all $s \in S$ to $a$

    \\ Aggregator Side

    Each cluster $u \in \mathcal{U}$ contains a group of models of same architectures.

    **for all** $u \in \mathcal{U}$ **do**

        $W_g^u \leftarrow$ Aggregate homogeneous models, for all $W_n \in u$, according to Equation 5

    **for all** $u \in \mathcal{U}$ **do**

        **for all** $W_n \in u$ **do**

            $W_n \leftarrow W_g^u$ \\ Fork $W_g^u$ into local models

    Send back updated models $W_s$ for all $s \in S$

---

### A.3.3 Decentralized Partial Training

Similar to DFML and decentralized FedAvg, in decentralized partial training methods each client owns a local model, and in each communication round several clients are randomly selected. One client is designated as the aggregator, while the others act as senders. The senders transmit their models to the aggregator after training their models locally. At the aggregator, the largest available model serves as the global model $W_g$. Algorithm 3 provides a detailed description of decentralized Federated Dropout, HeteroFL, and FedRolex. Model parameters in partial training methods are aggregated as follows:

$$W_{g,[i,j]} = \frac{1}{\sum_{n \in \mathcal{P}} d_n} \sum_{n \in \mathcal{P}} d_n W_{n,[i,j]}, \tag{6}$$

where $W_{[i,j]}$ is the $j^{th}$ parameter at layer $i$ of global model $W_g$, while $W_{n,[i,j]}$ is the $j^{th}$ parameter at layer $i$ of client $n$. The client weight is equal for all clients, $d_n = 1/|\mathcal{P}|$.

Decentralized Federated Dropout In each communication round $t$ within Federated Dropout (Caldas et al., 2018), sub-models (local models) are extracted from the centralized $W_g$ based on a random selection process. The parameters representing the sub-models that are randomly chosen through this selection are then transmitted to clients for local training. After local training, the updated parameters are sent back to $W_g$ for aggregation. The random extraction scheme for layer $i$ of sub-model $W_n$ for client $n$ is extracted from $W_g$ as follows:

$$\mathcal{X}_{n,i} = \{j_s \mid \text{integer } j_s \in [0, J_i - 1] \text{ for } 1 \leq s \leq \lfloor r_n J_i \rfloor\}, \tag{7}$$

where $\mathcal{X}_{n,i}$ is the parameter indices of layer $i$ extracted from $W_g$. $r_n$ denotes $W_n$ rate relative to $W_g$. $J_i$ denotes the total number of parameters in layer $i$ of $W_g$. A total of $\lfloor r_n J_i \rfloor$ is randomly selected from layer $i$ of $W_g$ for $W_n$.

Our derived decentralized version of Federated Dropout involves generating random indices (Equation 7) at the aggregator, guided by the largest available model ($W_g$). Subsequently, each model is assigned a random set of indices equivalent to its size. The aggregation process is then carried out using these randomly selected

---

**Algorithm 3** Decentralized ==Federated Dropout== , ==HeteroFL== , and ==FedRolex==

---

**Input:** Initialize $N$ clients, each client $n$ has a model $W_n$ and data $D_n$. Each model has a rate $r_n$, which is the model's size rate compared to the largest model in the network. Models with the same rate have the same initialization.

**for** communication round $t = 1, 2, ..., T$ **do**

    Randomly select one aggregator $a \in \{1, ..., N\}$

    Randomly select senders $S \subset \{1, ..., N\}$, $a \notin S$

    Participants $\mathcal{P} = S \cup \{a\}$

    \\ Client Side

    **for all** $n \in \mathcal{P}$ **do**

        **for all** batch $X_n \in$ local data $D_n$ **do**

            $W_n \leftarrow$ locally train $W_n$ using Equation 2

    Send locally updated models $W_s$ for all $s \in S$ to $a$

    \\ Aggregator Side

    Set $W_g$ to be like the largest available model

    Local models are assigned indices $\mathcal{X}_{n,i}$ for all $i$ and $n \in \mathcal{P}$, where $\mathcal{X}_{n,i}$ is from ==Equation 7== or ==Equation 8== or ==Equation 9==

    **for all** $n \in \mathcal{P}$ **do**

        $W_n \leftarrow W_{g,\mathcal{X}_{n,i}}$ for all $i$   \\ Split $W_g$ into local models

    Send back updated models $W_s$ for all $s \in S$

---

indices. Once the aggregation is finalized, sub-models are created from $W_g$ using the same set of indices. Finally, these sub-models are transmitted back to the respective participating clients.

Decentralized HeteroFL Unlike Federated Dropout, HeteroFL (Diao et al., 2020) consistently extracts sub-models from a predefined section of $W_g$. Specifically, HeteroFL extracts sub-models from $W_g$ starting from index 0 up to the maximum layer size of $W_n$. The extraction scheme is defined as follows:

$$\mathcal{X}_{n,i} = \{0, 1, ... \lfloor r_n J_i \rfloor - 1\}, \tag{8}$$

In the decentralized HeteroFL approach, the parameters of each layer in all sub-models share the same starting point (index 0). Consequently, at the aggregator, parameter averaging takes place with overlapping indices from the available models. After aggregation, the updated parameters of each layer from all models, spanning from index 0 up to the maximum size, are communicated back to their respective clients.

Decentralized FedRolex In FedRolex (Alam et al., 2022), local clients are initially generated from the global model beginning from index 0 and extending up to the local models' capacity. In the first communication round, the sub-models are generated similarly to HeteroFL. However, in each subsequent communication round, the starting point of the indices shifts to the right. The sub-model extraction in FedRolex is defined as follows:

$$\mathcal{X}_{n,i}^t = \begin{cases} \{\tilde{t}, \tilde{t}+1, ..., \tilde{t}+\lfloor r_n J_i \rfloor - 1\} & \text{if } \tilde{t} + \lfloor r_n J_i \rfloor \leq J_i, \\ \{\tilde{t}, \tilde{t}+1, ..., J_i - 1\} \cup \{0, 1, ..., \tilde{t}+\lfloor r_n J_i \rfloor - 1 - J_i\} & \text{otherwise,} \end{cases} \tag{9}$$

where $\tilde{t} = t \mod J_i$. $t$ is the current communication round and $J_i$ is the size of layer $i$ of $W_g$.

In decentralized FedRolex, the size of $W_g$ is determined by the largest available model, and as a result, the rightward shift in indices is computed based on the current communication round $t$ and $J_i$ of the selected $W_g$. The indices are calculated using $\mathcal{X}_{n,i}^t$ from Equation 9. These indices are utilized for aggregation and to extract the updated local models after aggregation. In decentralized FedRolex, since aggregators must be aware of $t$ to compute the indices, each aggregator needs to communicate $t$ to the next aggregator.

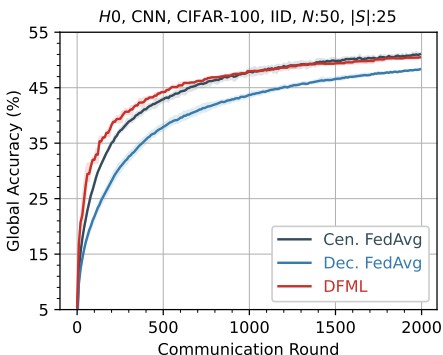 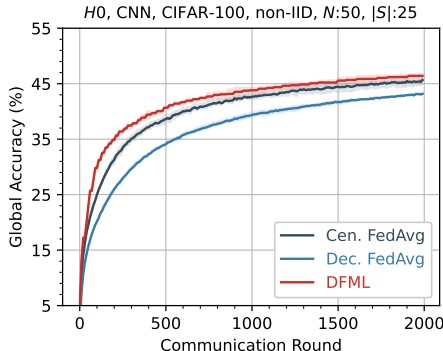

Figure 12: Performance comparison between centralized (with server) and decentralized (serverless) settings.

### A.3.4 Decentralized FML

FML (Shen et al., 2020) is a centralized framework that relies on a central server. In FML, each client owns a local model, which is not transmitted during training. In each communication round, the global model forks its model into all participating clients. Shen et al. 2020 named the forked models: *meme models*. Subsequently, at each client, the local (heterogeneous) model engages in mutual learning with the meme (homogeneous) model using their respective local data. After mutual learning is complete, the meme models are communicated back to the server for aggregation.

In decentralized FML, two models are dedicated to each client: the first is the heterogeneous model, and the second is the homogeneous (meme) model. In each communication round, several clients perform mutual learning between their heterogeneous and homogeneous models using their local data. Next, the homogeneous models from all participating clients are transmitted to the aggregator for aggregation. After aggregation is complete, the aggregated model is transmitted back to all participating clients. This process repeats for the remaining communication rounds.

We decentralized FML for comparison with our proposed DFML. **It is crucial to emphasize that decentralized FML and our proposed DFML are two distinct frameworks.** The key differences are as follows: 1) DFML uses one heterogeneous model per client for training, while decentralized FML uses two models per client for training; 2) DFML aims to transfer global knowledge to heterogeneous models, whereas FML treats the heterogeneous models as personalized models and the homogeneous models to hold the global knowledge; and 3) FML requires a server, and we derived the decentralized version of FML to facilitate comparison with our proposed DFML.

### A.4 DFML: Further Analysis

### A.4.1 Centralized vs Decentralized

We highlight a challenge of a serverless setting compared to having a centralized server in the network. For simplicity, we illustrate the difference using FedAvg. In centralized FedAvg, a global model exists at the server, and in each communication round, the global model is distributed to randomly selected clients. Even when partial client participation, there is always one version of the global model at the server. In a decentralized setting with partial participation, clients send their model to another client for aggregation and receive back the aggregated model, resulting in a version of the global model that differs from the previous round especially with different clients selected for participation. The existence of multiple versions of global models among clients in a serverless network affects the convergence speed compared to a centralized server where the latest version of the global model is transmitted to clients in every round. Figure 12 illustrates the convergence speed drop between a centralized and decentralized network using FedAvg and homogeneous architectures. Additionally, our proposed DFML challenges the centralized FedAvg. As mentioned in Section 1, there are other advantages of a serverless network compared to having a centralized server.

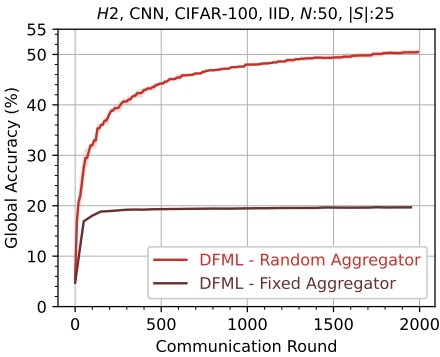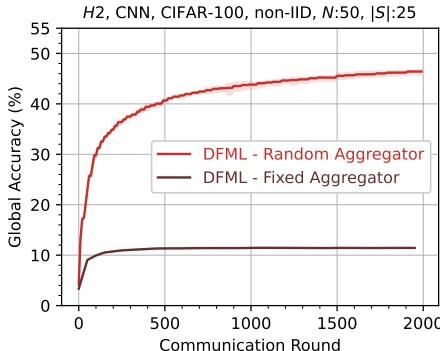

Figure 13: Performance comparison between using the same client (client 0) and randomly selecting a client in each communication round to perform the aggregation.

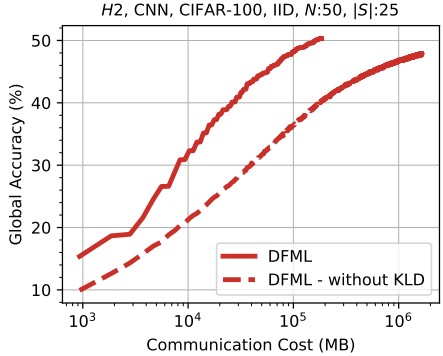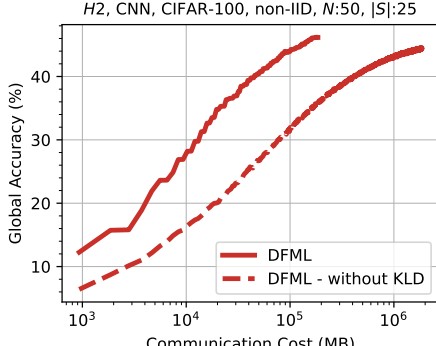

Figure 14: Performance comparison between using both distillation and supervision signals and only using the supervision signal at the aggregator to perform knowledge transfer.

### A.4.2 Fixed vs Random Aggregator

Figure 13 shows that using a fixed client to perform the aggregation significantly limits performance since the same local data is used in the knowledge transfer between models. In this experiment, the senders are randomly selected in each communication round. while the aggregator is always client 0.

### A.4.3 Effect of Knowledge Distillation

We explore the effect of utilizing a distillation signal in the objective function compared to just having a supervision signal at the aggregator (Equation 1). Figure 14 illustrates the communication cost difference between having both supervision and distillation signals in the aggregator's objective function versus just having the supervision signal. The figure shows that without the distillation signal, the computation cost is approximately an order of magnitude higher than with the addition of the distillation signal to the supervision signal at the aggregator, as described in Equation 1.

### A.4.4 Convergence Speedup

We illustrate in Figures 15, 16, and 17 the convergence speedup achieved by DFML compared to the baselines under three heterogeneity settings: homogeneous architectures, restrictive heterogeneous architectures, and nonrestrictive heterogeneous architectures; respectively.

### A.4.5 Local Accuracy

Our proposed DFML not only surpasses the baselines in global accuracy but also achieves competitive results in local accuracy. As shown in Figure 18, the local accuracy attained by DFML generally exceeds

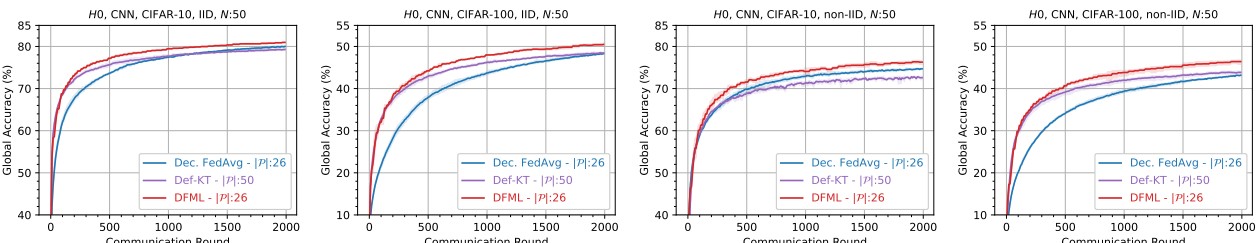

Figure 15: Comparison between DFML, decentralized FedAvg, and Def-KT using homogeneous architectures.

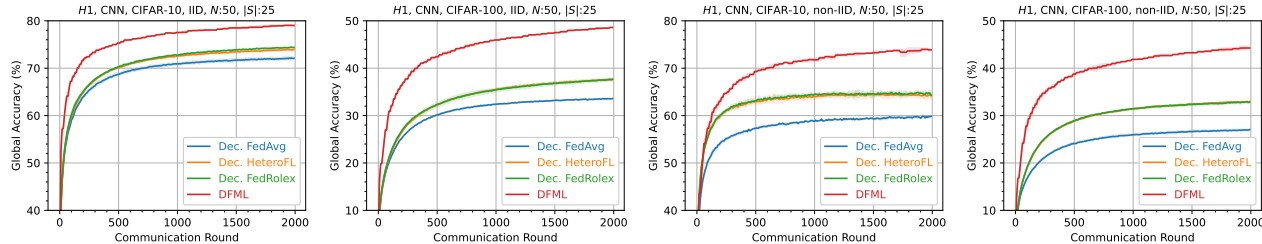

Figure 16: Comparison between DFML, decentralized partial training algorithms, and decentralized FedAvg using restrictive heterogeneous architectures.

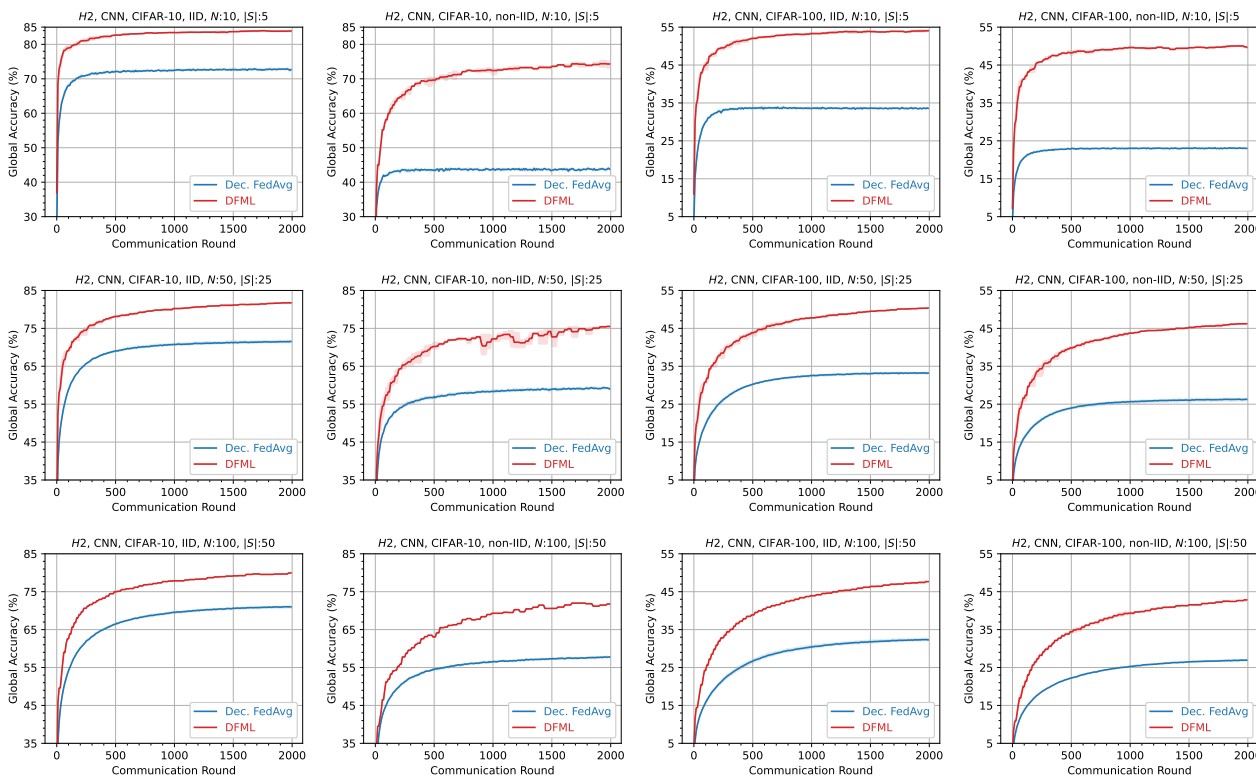

Figure 17: Comparison between DFML and decentralized FedAvg for different numbers of clients and data distributions using non-restrictive heterogeneous architectures.

that of decentralized FedAvg. Although DFML exhibits slightly lower local accuracy compared to decentralized FedAvg in the CIFAR-10 non-IID experiment with 10 clients, it remains competitive. Moreover, the corresponding global accuracy achieved by DFML in that experiment surpasses decentralized FedAvg.

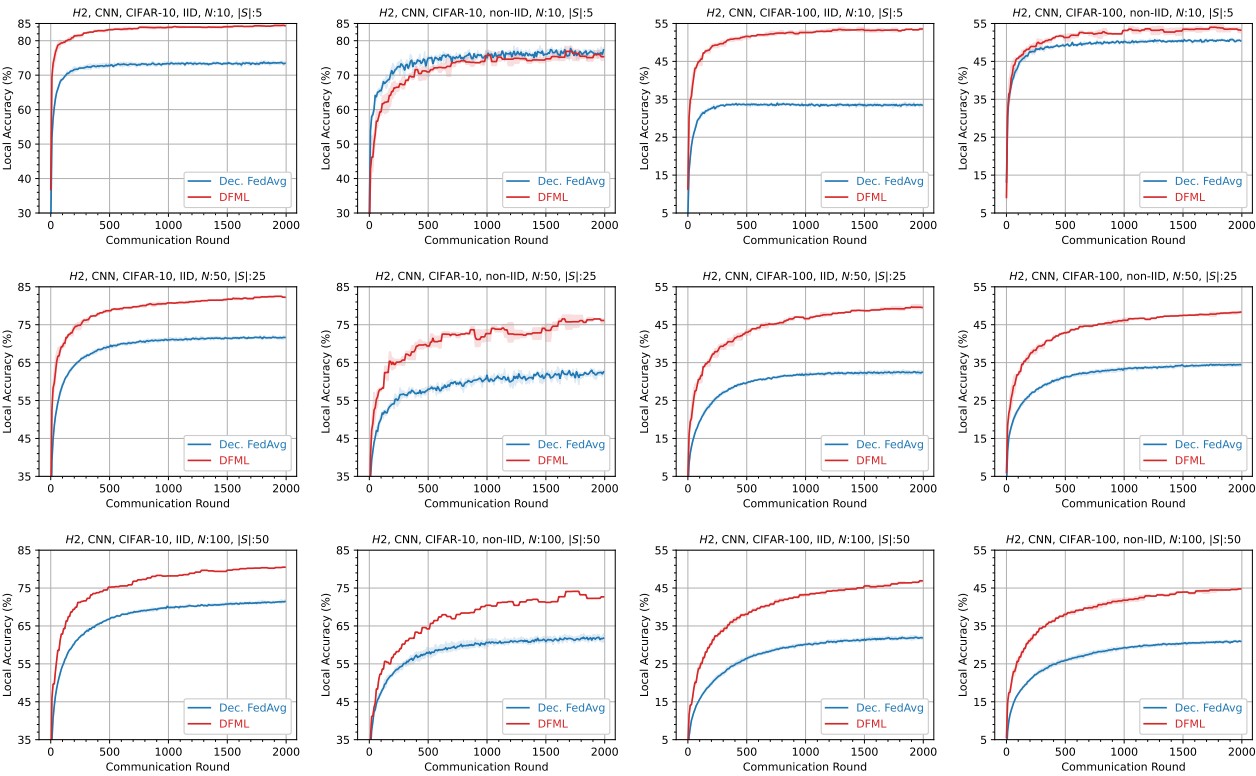

Figure 18: Local accuracy comparison between DFML and decentralized FedAvg for different numbers of clients and data distributions using non-restrictive heterogeneous architectures.

### A.4.6 Different Number of Participants

We investigate the performance of DFML with a varying number of senders $S$ in each communication round. Figure 19 compares DFML and decentralized FedAvg with 50%, 20%, and 10% senders. DFML exhibits effective learning even with fewer participants compared to decentralized FedAvg. Additionally, both methods with a reduced number of participants demonstrate slower convergence speed compared to the 50% scenario, which is expected.

Due to the low participation rate, we increase the number of peak updates. With a limited number of participating models, updating them only when the maximum $\alpha$ is reached results in slower convergence speeds. Therefore, we allow multiple updates instead of updating the models solely at the maximum $\alpha$. We estimate that an appropriate number of peak updates is $\frac{N}{|S|}$ for $|S| < 50\%$ of the clients. Consequently, with $|S| = 10\%$, updates are applied in the largest five $\alpha$ values. If cyclical $\alpha$ is not added to DFML, adjusting the number of peak updates is unnecessary, as peak models will no longer be needed and only the updated models must be communicated back to senders.

### A.4.7 Weighted vs Vanilla Average of KL Divergences (KLs)

In Equation 3, we use a weighted average of KL divergences (KLs) between all teacher models and the student. The weighting is determined based on the number of trainable parameters in each teacher model. Figure 20 demonstrates that the weighted average leads to better convergence speed and global accuracy compared to vanilla averaging. The reason is that larger models tend to have a higher probability of possessing finer knowledge, thus giving them more weight during knowledge distillation results in better knowledge transfer.

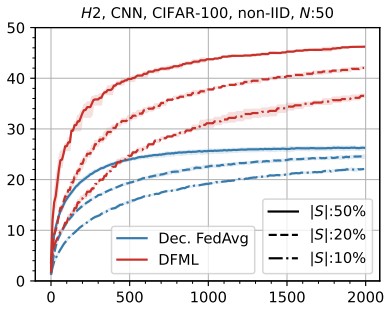 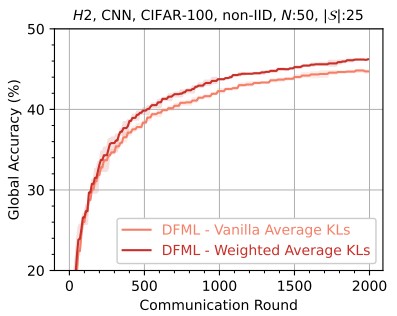 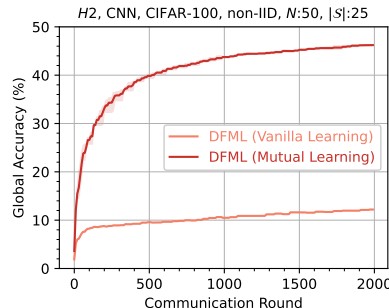

Figure 19: Evaluating DFML against decentralized FedAvg with fewer number of senders $S$.

Figure 20: Weighted vs vanilla average of KL divergences (KLs) in the distillation signal of the objective function.

Figure 21: Mutual vs vanilla knowledge transfer.

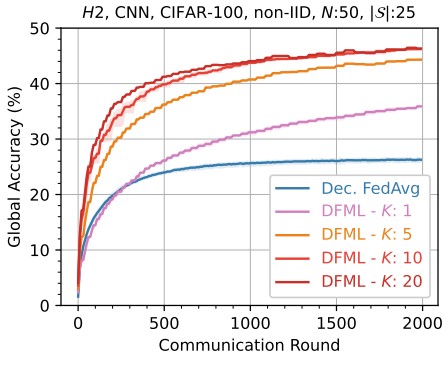 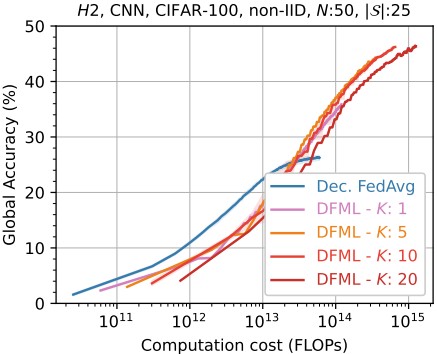

Figure 22: Effect of increasing mutual learning epochs $K$ on global accuracy with respect to (Left) communication rounds and (Right) computation cost.

### A.4.8 Mutual vs Vanilla Knowledge Transfer

With vanilla knowledge transfer, only the aggregator's model is updated, with the other models act as teachers. Conversely, with mutual learning, all models are updated. Figure 21 demonstrates the significant improvement in global accuracy and convergence speed achieved through mutual learning compared to vanilla knowledge transfer.

### A.4.9 Effect of Increasing Mutual Learning Epochs $K$

Figure 22 shows that increasing the number of mutual learning epochs $K$ at each aggregator contributes to a faster convergence speed. A higher number of $K$ enables more knowledge to be distilled from the experts, leading to improved convergence speed. This, in turn, results in a more efficient communication cost throughout training. On the other hand, the computational cost will increase significantly. From the figure we can see that increasing the $K$ beyond 10 does not result in significant enhancement in convergence speed at the expense of adding more communications between clients. Further, selecting $K$ lower than 10 will save computation but will increase the communication cost. Thus, there is a trade-off between computation and communication cost for convergence. Last, Even though decentralize FedAvg is more computation efficient than DFML, but it does not reach to the same global accuracy level as DFML.

### A.4.10 Different Topology

In all previous experiments, we used a mesh topology, where all clients can reach each other. However, in this subsection, we explore a different network topology, as illustrated in Figure 23. In this new topology, clients are divided into two groups, each connected in a mesh configuration, with a single link connecting

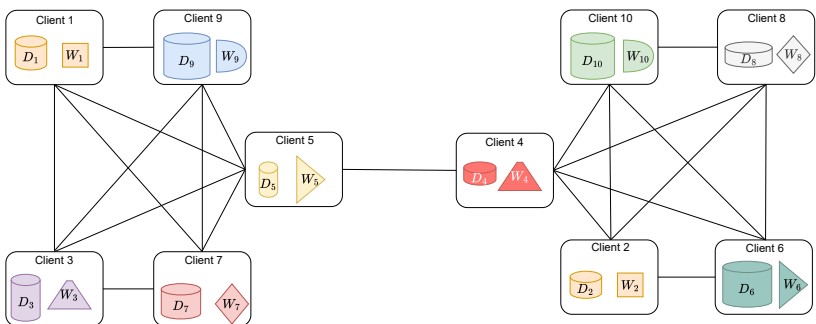

Figure 23: Different Topology.

Table 11: Global accuracy comparison using restrictive heterogeneous architectures with 50 clients and 25 senders. The experiments are conducted using ResNet architectures using the topology illustrated in Figure 23.

| | CIFAR-10 | | CIFAR-100 | |
|---|---|---|---|---|
| **Method** | **IID** | **non-IID** | **IID** | **non-IID** |
| Dec. FedAvg | $44.71_{\pm 0.12}$ | $22.11_{\pm 0.06}$ | $10.36_{\pm 0.16}$ | $7.51_{\pm 0.14}$ |
| Dec. HeteroFL | $58.91_{\pm 0.17}$ | $43.45_{\pm 0.21}$ | $18.24_{\pm 0.13}$ | $14.81_{\pm 0.22}$ |
| Dec. FedRolex | $58.91_{\pm 0.09}$ | $43.54_{\pm 0.19}$ | $18.20_{\pm 0.08}$ | $15.10_{\pm 0.11}$ |
| **DFML (Ours)** | $\mathbf{78.87}_{\pm 0.01}$ | $\mathbf{68.52}_{\pm 0.02}$ | $\mathbf{45.02}_{\pm 0.02}$ | $\mathbf{40.31}_{\pm 0.03}$ |

Table 12: Global accuracy comparison using nonrestrictive heterogeneous architectures with 50 clients and 25 senders. The experiments are conducted using CNN architectures using the topology illustrated in Figure 23.

| | CIFAR-10 | | CIFAR-100 | |
|---|---|---|---|---|
| **Method** | **IID** | **non-IID** | **IID** | **non-IID** |
| Dec. FedAvg | $51.45_{\pm 0.01}$ | $25.03_{\pm 0.02}$ | $27.10_{\pm 0.11}$ | $20.16_{\pm 0.21}$ |
| **DFML (Ours)** | $\mathbf{78.87}_{\pm 0.10}$ | $\mathbf{68.52}_{\pm 0.18}$ | $\mathbf{45.01}_{\pm 0.15}$ | $\mathbf{40.30}_{\pm 0.15}$ |

the two groups. The first group contains clients with odd addresses $[1, 3, 5, \ldots, N-1]$ and the second group contains clients with even addresses $[2, 4, 6, \ldots, N]$. The clients connecting group 1 and group 2 have the "median" address from the list of addresses in its group in their respective groups.

We conducted experiments using both restrictive and nonrestrictive heterogeneous architectures. The results, provided in Tables 11 and 12, demonstrates that even with this different topology, DFML continues to surpass the baselines.

### A.4.11 DFML vs Decentralized FML

In decentralized FML, the clients' heterogeneous models are the same as the models used in our proposed DFML, which are based on CNN architectures and mode $H2$ from Table 10. The homogeneous model size used is $[32, 64]$, which is the median of models of the five different architectures. Figure 24 compares our DFML and decentralized FML. We include the global accuracy of the local heterogeneous and the homogeneous meme models. We observe that the homogeneous meme models have a better convergence speed than our DFML, and in some cases, lead to better final accuracy. However, the global knowledge performance of heterogeneous models in decentralized FML is much worse than DFML and is even lower than decentralized FedAvg. As our goal is to transfer global knowledge to clients' heterogeneous models, we consider that our DFML significantly outperforms the decentralized FML framework.

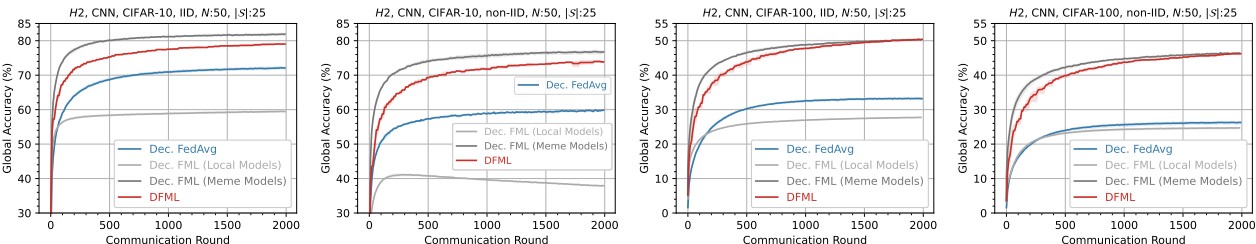

Figure 24: Comparison between our proposed DFML and decentralized FML.

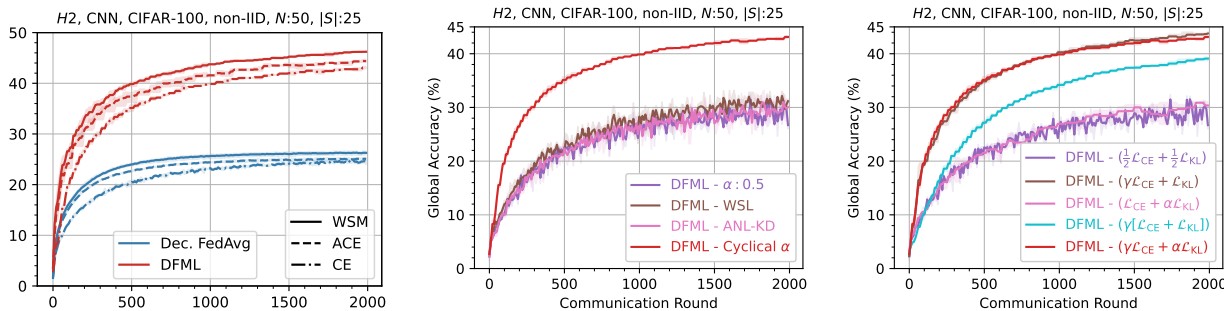

Figure 25: Comparison between different supervision signals.

Figure 26: Comparing various adaptive techniques with our proposed DFML. The supervision signal used is CE.

Figure 27: Different schedulers assigned to the loss components in the objective function.

## A.5 Cyclical Knowledge Distillation: Further Analysis

### A.5.1 Different Supervision Signals

As previously mentioned, mitigating catastrophic forgetting is crucial in applications such as FL where data distribution shift exists across clients. Training models on one dataset can lead to optimizing its objective towards that specific local data, causing it to forget tasks learned previously from other clients. ACE (Caccia et al., 2021) and WSM (Legate et al., 2023) are two approaches to mitigate catastrophic forgetting. Figure 25 illustrates the improvement in global accuracy when the supervision signal is changed from CE to ACE or WSM. Results show that WSM as a supervision signal leads to the highest global accuracy, as it takes into account the number of samples for each class label in the clients. The maximum range of $\alpha$ oscillation is tuned for each supervision signal to achieve the best final accuracy. Tuning the maximum $\alpha$ range is important, particularly in scenarios where the supervisory signal is non-noisy, such as in IID distribution shift, or when using ACE or WSM in non-IID settings. Completely diminishing the supervisory signal (equivalent to setting $\alpha = 1$) in such cases would lead to a performance decline. Therefore, in situation where the supervision signal is not noisy, the maximum $\alpha$ value is better to be set to 0.8 or 0.9. For instance, in non-IID cases with CE as the supervision signal, where the signal is very noisy, setting $\alpha = 1$ yields the best performance.

### A.5.2 Different Adaptive Techniques

In Figure 26, we compare the performance of different adaptive techniques, including WSL (Zhou et al., 2021) and ANL-KD (Clark et al., 2019), with our proposed cyclical DFML framework. The results indicate that WSL and ANL-KD show negligible improvement compared to DFML with a fixed $\alpha = 0.5$.

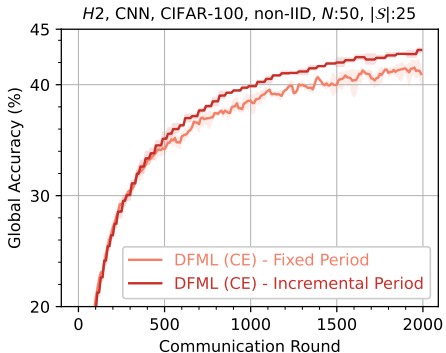

Figure 28: Comparison between fixing and increasing the period throughout training. The initial period is set to 10.

### A.5.3 Different Schedulers for Loss Components

In Equation 1, the objective function consists of two loss components: supervision and distillation loss signals. In Equation 10, we examine the impact of independently varying each loss component on global accuracy. Specifically, we compare additional scenarios where one scheduler is assigned to $\mathcal{L}_{\mathrm{CE}}$ alone, another scenario where a scheduler is assigned to $\mathcal{L}_{\mathrm{KL}}$ alone, and a third scenario where one scheduler scales both loss components with the same factor. For this experiment, we use CE as the supervision signal, as the improvements are more notable under the non-IID distribution shifts. Figure 27 illustrates the comparison of assigning independent schedulers to loss components. In this experiment, $\gamma$ oscillates from $1 \to 0$, and $\alpha$ oscillates from $0 \to 1$.

$$\mathcal{L} = \gamma \mathcal{L}_{\mathrm{CE}} + \alpha \mathcal{L}_{\mathrm{KL}} \tag{10}$$

Scaling the distillation signal alone and leaving the scale of $\mathcal{L}_{\mathrm{CE}}$ untouched does not yield any advancement in performance compared to keeping $\alpha$ fixed for both loss components. This indicates that the supervision signal has a more significant impact than the distillation signal. Performance gains are observed when the $\mathcal{L}_{\mathrm{CE}}$ signal is reduced, allowing more influence on the $\mathcal{L}_{\mathrm{KL}}$ signal. In contrast, when $\mathcal{L}_{\mathrm{CE}}$ oscillates while the $\mathcal{L}_{\mathrm{KL}}$ scale is kept fixed, it results in the same performance as when both $\mathcal{L}_{\mathrm{CE}}$ and $\mathcal{L}_{\mathrm{KL}}$ signals are scaled in opposite directions (Equation 1). This is because the $\mathcal{L}_{\mathrm{CE}}$ signal is dominant without any scaling, and as it diminishes it allows the $\mathcal{L}_{\mathrm{KL}}$ signal to take precedence. The peak value is attained when the $\mathcal{L}_{\mathrm{CE}}$ signal reaches 0, and the $\mathcal{L}_{\mathrm{KL}}$ signal's scale is 1. Finally, scaling both $\mathcal{L}_{\mathrm{CE}}$ and $\mathcal{L}_{\mathrm{KL}}$ signals with a common scheduler leads to inferior performance compared to scaling $\mathcal{L}_{\mathrm{CE}}$ alone or scaling both signals in opposite directions. The poor performance of oscillating $\mathcal{L}_{\mathrm{KL}}$ signal alone is attributed to the continuous dominance of $\mathcal{L}_{\mathrm{CE}}$ signal during mutual learning, causing the models to drift toward the aggregator's local data.

### A.5.4 Fixed vs Increasing Period

Figure 28 demonstrates that increasing the period over time results in better convergence speed, higher global accuracy, and enhanced stability. The period is initially set to 10. In the fixed period experiment, the period is kept constant, while in the increasing period experiment, the period is incremented. Starting with a small period is crucial for more frequent peak updates, which accelerates convergence speed. However, over time, increasing the period proves beneficial, allowing models to transition from the supervision to the distillation signal more slowly. This extended time in the distillation-dominant region enhances global accuracy. For instance, if all clients are participating and the period is initially set at 100, better accuracy is achieved after 100 communication rounds compared to a constant period of 10. However, the convergence speed is notably affected, as a period of 100, results in one peak at communication round 100, while a period of 10 leads to 10 peaks. Further, in scenarios with partial participation, then after 100 rounds only the participating clients will be updated. Whereas a smaller initial period ensures that, on average, all clients are updated several times within the first 100 rounds. Therefore, to reap the benefits of high convergence speed and improved final accuracy, we set the period to be initially small and increment it gradually.

