# OpenReview forum: "DFML: Decentralized Federated Mutual Learning"
_TMLR — Accepted by TMLR_

### Review · Reviewer_LN2c · 2024-05-10

**Summary Of Contributions:**

This work aims at serverless and relaxed-constraint FL. To this end, the authors propose a novel learning framework in which models are uploaded to certain clients and fine-tuned using their data. Some experiments are conducted to verify the effectiveness of the proposed method.

**Audience:**

Yes

**Claims And Evidence:**

No

**Requested Changes:**

1 The motivation/significance of the highlighted contributions should be carefully clarified, see Weaknesses 1, 2, and 5.

2 More experiments are required.

**Strengths And Weaknesses:**

[Strengths ]

1 The studied problem is promising.

2 Introducing accuracy on IID data is convincing.

3 Multiple architectures are leveraged for evaluation.


[Weaknesses]

1 The claimed "no server"  (or serverless) is confusing and requires careful clarification. Specifically, it seems like changing the server in FedAvg to the "no server" setting claimed in this work induces no challenge. Namely, we can perform aggregation on any client, as claimed in this work.

2 The claimed architectural constraints are widely studied in one-shot FL, where limited or no constraints are leveraged.

3 The proposed cyclic knowledge distillation would introduce numerous costs as the FL system is scaling up, e.g., with more data or clients.

4 The experimental results are not convincing, as merely CIFAR10 and CIFAR100 are considered in the evaluation.

5 The proposed method would approach the centralized training setting and performance if the proposed learning paradigm does not fix the client for knowledge distillation.

---

> ### Author Response · Authors · 2024-05-27
>
> Dear Reviewer LN2c,
>
> We thank you for taking the time to review our paper and provide valuable comments to help us clarify and improve our work. We have updated the manuscript based on your feedback, with the changes highlighted in blue. Here is our response:
>
> 1) The claimed "no server" (or serverless) is confusing and requires careful clarification. Specifically, it seems like changing the server in FedAvg to the "no server" setting claimed in this work induces no challenge. Namely, we can perform aggregation on any client, as claimed in this work.
>
> Response: We added a subsection on page 20 illustrating the challenges between centralized and decentralized settings. We would like to emphasize that our definition of no server, is to not have one "centralized" client acting as the aggregator. Using a centralized server for FedAvg means the server keeps track of the global model, sending it to randomly selected clients for local training in each round. The locally trained models are then aggregated at the server to update the global model, which is sent to another set of randomly selected clients in the next round. With a centralized server, there is always one version of the global model, which helps in convergence speed, as shown in Figure 12. In a serverless setting, there is no central place to keep track of the global model, resulting in several versions of the global model across the network based on client participation in each round. If a different set of participants is selected in consecutive rounds, there will be multiple versions of the global models. We consider partial participation since clients do not have the same computational resources as a server. The figure shows that centralized FedAvg has a higher convergence speed compared to decentralized FedAvg. Therefore, the serverless setting poses a challenge by affecting convergence speed. Additionally, other advantages of not relying on a centralized server includes robustness, avoiding a single point of failure, the need for constant server availability, and eliminating communication bottlenecks.
>
> 2) The claimed architectural constraints are widely studied in one-shot FL, where limited or no constraints are leveraged.
>
> Response: To the best of our knowledge, no work in FL that performs knowledge transfer between heterogeneous architectures (without any constraints) and without utilizing additional data. Although HeteroFL and FedRolex are centralized methods that impose constraints on architectures, we implemented decentralized versions of HeteroFL and FedRolex with the same architectural constraints to use them as baselines. The only degree of freedom is the width of each layer but the same model type must be used (e.g., ResNet). Results show that DFML outperforms decentralized HeteroFL and FedRolex. Additionally, FedMD requires public data to be available to perform knowledge transfer. Moreover, some FL works generate synthetic data to use for distill knowledge; however, this strategy introduce an extra privacy concern. We avoided this challenge of privacy concern and do not consider generating synthetic data. For fair comparison, we did not compare DFML with any method that utilize extra data.
>
> 3) The proposed cyclic knowledge distillation would introduce numerous costs as the FL system is scaling up, e.g., with more data or clients.
>
> Response: With cyclic knowledge distillation, only one parameter needs to be transferred between the aggregator and the client. Cyclic knowledge distillation does not incur additional communication or computation cost when the FL system is scaled up with more data or clients. Thus, the communication and computation costs remain constant due to the cyclic part. Results with different number of clients and data as provided in Table 8 and Figure 16.
>
> 4) The experimental results are not convincing, as merely CIFAR10 and CIFAR100 are considered in the evaluation.
>
> Response: We evaluated our proposed DFML with three additional datasets (Caltech101, OxfordPets, and StanfordCars). We also conducted experiments using restrictive and nonrestrictive heterogeneous architectures. The results are consistent with our previous findings, showing that DFML outperforms the baselines. The new results are presented in Tables 5 and 7.
>
> 5) The proposed method would approach the centralized training setting and performance if the proposed learning paradigm does not fix the client for knowledge distillation.
>
> Response: DFML chooses a different client every round to perform knowledge distillation. We added Figure 13 to show that fixing the client results in inferior performance. Additionally, we added Figure 14 to demonstrate that randomly choosing a client but only using the supervision signal without knowledge distillation severely affects convergence speed.
>
> Best regards,
> The authors.

---

> > ### Comment · Reviewer_LN2c · 2024-07-16
> > **Detailed comments**
> >
> > Thanks for the response.
> >
> > For A1:
> >
> > The detailed clarification raises a question: what is the difference between the claimed setting and the setting of the already existing Asynchronous Federated Learning? Moreover, could you highlight the motivation and provide more practical scenarios of this setting?
> >
> > For A2:
> >
> > Please see the papers below.
> >
> > [1] Practical one-shot federated learning for cross-silo setting. Li et al. IJCAI, 2021
> > [2] Dense: Data-free one-shot federated learning. Zhang et al. NeurIPS, 2022.
> > [3] Data-free one-shot federated learning under very high statistical heterogeneity. Heinbaugh et al. ICLR, 2023
> > [4] Enhancing One-Shot Federated Learning Through Data and Ensemble Co-Boosting. Dail et al. ICLR, 2024
> >
> > For A3:
> >
> > The results are confusing. If the authors prefer the model performance, they should highlight the performance gain. At the same time, the authors should report the communication round to attain that accuracy. However, the authors report the communication cost under a different scenario. This is not a good explanation.
> >
> > For A4:
> >
> > The experiments consider only image datasets. Moreover, this work lacks relatively large datasets, e.g., FMNIST and Tinyimagenet.
> >
> > For A5:
> >
> > "DFML chooses a different client every round to perform knowledge distillation." It seems a strong constraint for the FL system, which conflicts with the claimed setting.

---

> > > ### Comment · Action_Editor_hVSN · 2024-07-18
> > >
> > > Dear Authors,
> > >
> > > Could you reply to the questions?
> > >
> > > Best wishes,
> > > AE

---

> > > ### Author Response · Authors · 2024-07-18
> > >
> > > Dear Reviewer LN2c,
> > >
> > > We would like to extend our gratitude for your time and effort in reviewing our work.
> > >
> > > A1 Response: Our DFML is considered synchronous FL as the next aggregator is selected by the current aggregator upon completion of the aggregation process. Regarding practical scenarios of our setting: 1) 5G smart home devices and controllers with varying storage capacities connected via Wi-Fi Direct protocol to collaboratively learn how to serve in-home users; 2) Smartphones from different brands connected via a block-chain protocol to share their models without accessing the cloud. Both the motivation for using decentralized federated learning and practical scenarios can be found in the Introduction section.
> > >
> > > A2 Response: In the Introduction section we have cited the above papers and noted that the key difference between our works and theirs is that they require generating synthetic data, which may compromise privacy. In our proposed DFML work, we do not require any auxiliary data (public or synthetic). DFML depends solely on existing clients' local data for knowledge transfer.
> > >
> > > A3 Response: We meant to refer you to Table 9 and Figure 17. The numbers were shifted due to the addition of figures and tables based on the reviewers' requests. Please refer to Table 9 for an ablation on the effect of cyclical alpha on DFML, and refer to Figure 17 for the convergence speed results, with the x-axis representing communication rounds.
> > >
> > > A4 Response: Indeed, so far we have studied DFML with only image datasets. In the extended works of DFML, we will consider different data modalities. Furthermore, in Table 5, we added results of DFML using FMNIST dataset as requested.
> > >
> > > A5 Response: We would like to emphasize that the aggregator node can be different every round. If we were to use the same client every round, the scenario would degenerate to centralized FL. Please refer to Figure 13, where we have provided results for using a fixed node as the aggregator versus using a random node in every round. In decentralized FL, clients come and go, and thus we can not rely on a fixed client to be the aggregator.

---

> > > > ### Comment · Reviewer_LN2c · 2024-08-06
> > > > **Thanks for the response**
> > > >
> > > > The outstanding responses have addressed my concerns. Accordingly, I am raising the score. BTW, sorry for the late comments, you know, I am also an author :)

---

> > > > > ### Author Response · Authors · 2024-08-06
> > > > >
> > > > > Thank you for raising the score. We appreciate the time you took to review our work. Your feedback has been invaluable in improving the quality of our submission.

---

> ### Comment · Action_Editor_hVSN · 2024-07-12
>
> Your rebuttal is not visible to the reviewer. Please revise.

---

> > ### Author Response · Authors · 2024-07-15
> >
> > The visibility of the rebuttal has been revised. Thanks for letting us know.

---

### Review · Reviewer_Pqap · 2024-06-03

**Summary Of Contributions:**

This paper presents Decentralized Federated Mutual Learning (DFML) to address heterogeneity in a Decentralized Federated Learning (DFL) setting. Specifically, DFML transmits models to a randomly selected aggregator to handle model heterogeneity and uses re-weighted SoftMax to address data heterogeneity. The aggregation is performed by distilling knowledge on the aggregator client. A cyclical approach is used to balance two loss functions. The experimental results show the proposed method is effective in addressing the non-IID challenge.

**Audience:**

Yes

**Claims And Evidence:**

Yes

**Requested Changes:**

See above.

Please consider using \citep{} if the authors are not part of the sentence.

**Strengths And Weaknesses:**

Strengths:

Handling heterogeneity in Decentralized Federated Learning setting is well-motivated.

The experiments are extensive and easy to follow.

The results show the proposed method is effective.

Weaknesses:

Although the communication cost is not impacted, the computation cost might be high for the selected aggregator (client).

If a less powerful edge device is selected as the aggregator and it cannot run the model sent from other clients, how can the method still work? For example, if an edge device can only deploy a CNN model, how can it aggregate a ViT model sent from other clients?

Will the learned model bias toward the data on the last aggregator at the end of training?

In the global accuracy comparison using homogeneous CNN architectures, can the authors compare DFML with a decentralized version of an FL method designed for data heterogeneity?

---

> ### Author Response · Authors · 2024-06-11
>
> Dear Reviewer Pqap,
>
> Thank you for your interest in our work and for providing valuable comments that have helped us clarify and enhance our research. We have carefully considered your feedback and revised the manuscript accordingly. The changes have been highlighted in red for your convenience. Below, we provide a detailed response to your comments and outline the specific modifications made to the manuscript:
>
> 1) Although the communication cost is not impacted, the computation cost might be high for the selected aggregator (client). If a less powerful edge device is selected as the aggregator and it cannot run the model sent from other clients, how can the method still work? For example, if an edge device can only deploy a CNN model, how can it aggregate a ViT model sent from other clients?
>
> Response: The reviewer has raised a valid point. In this research work, we did not explore the computational limitations of clients. We assumed a minimal computational requirement within the network, where each client has the capability to load and update any other model in the network. In future work, we will consider routing senders only to aggregators with sufficient computational power to handle the senders' models. Furthermore, even if we constrain aggregators to handle computation up to a certain limit, our proposed DFML will still outperform decentralized FedAvg. For example, if we restrict aggregators to receive networks of only the same model sizes or smaller (an aggregator with a ViT model should only receive ViT models or smaller ones like CNNs), FedAvg will only aggregate models of same architecture. In the other scenario, where the aggregator has a CNN model deployed and should only receive CNN model, DFML still outperforms FedAvg as shown in our homogeneous results. In summary, although our current results does not account for computational constraints, our proposed method should still outperform decentralized FedAvg, even if such constraints existed.
>
> 2) Will the learned model bias toward the data on the last aggregator at the end of training? In the global accuracy comparison using homogeneous CNN architectures, can the authors compare DFML with a decentralized version of an FL method designed for data heterogeneity?
>
> Response: With DFML, the learned models do not drift toward the data of the last aggregator because WSM is used as the supervision signal instead of CE. Table 9 compares DFML with CE and WSM supervision signals, showing that Without cyclical $\alpha$, the results with non-IID data are significantly lower compared to when WSM is used. Additionally, all our baselines, including decentralized FedAvg, are implemented with WSM to prevent them from drifting due to heavy data heterogeneity and to allow a fair comparison with DFML. Furthermore, we implemented a decentralized version of FedProx, and the results are presented in Table 3. FedProx is designed to handle data heterogeneity by preventing the locally trained model from deviating significantly from the global model. However, the results show that decentralized FedProx yielded approximately the same results as decentralized FedAvg. In centralized FedProx, the same global model is used in the partially selected clients as a reference to prevent local models from deviating. In contrast, decentralized FedProx lacks a single global model; each client has its version of the global model (its local model). Therefore, we use a copy of the client's model as the global model, aggregated with other models from previous rounds. Implementation details of decentralized FedProx are provided in Appendix A.3.2.
>
> 3) Please consider using \citep{} if the authors are not part of the sentence.
>
> Response: The manuscript has been updated based on the given feedback.

---

> > ### Comment · Reviewer_Pqap · 2024-07-16
> >
> > Thanks for your response.
> >
> > 1. I noticed that the revised version includes the memory limitation. It is acceptable to introduce this condition if explicitly stated.  I am also interested in whether the aggregated model would be biased toward those chosen clients if routing senders only to aggregators with sufficient computational power to handle the senders' models. Maybe you can explore it further in your future work.
> >
> > 2. The choice of hyperparameter $\mu$ is missing in the revised manuscript. $\mu$ is an important hyperparameter and needs to be carefully tuned for different experiments in my experience.

---

> > > ### Author Response · Authors · 2024-07-18
> > >
> > > Dear Reviewer Pqap,
> > >
> > > We would like to thank you again for your time and effort in reviewing our work.
> > >
> > > A1 Response: That is a great suggestion! We will explore this further in our future work.
> > >
> > > A2 Response: The manuscript has been updated accordingly to include the values used to tune the hyperparameter $\mu$.

---

### Review · Reviewer_TnFy · 2024-06-11

**Summary Of Contributions:**

The paper proposes a novel Decentralized Federated Mutual Learning (DFML) framework that is serverless, supports heterogeneous models without architectural constraints, and does not rely on additional data. DFML handles model and data heterogeneity through mutual learning that distills knowledge between clients, and by cyclically varying the amount of supervision and distillation signals. Extensive experiments demonstrate DFML consistently outperforms prevalent baselines in convergence speed and global accuracy under various conditions.

**Audience:**

Yes

**Claims And Evidence:**

Yes

**Requested Changes:**

Overall, I think this paper is well-written and has significant reasearch value. However, there are some issues to be addressed.

- (major) Investigate the impact of different network topologies on the performance of DFML.
- (major) Conduct an ablation study to evaluate the individual contributions of the key components of DFML, such as mutual learning, cyclical variation of α, and the use of peak models.
- (minor) Expand the experiments to cover a wider range of real-world datasets and applications.
- (minor) Include a theoretical analysis of the convergence properties and limitations of DFML.

**Strengths And Weaknesses:**

Strengths
- DFML supports nonrestrictive heterogeneous models and does not rely on additional data, making it more practical for real-world scenarios.
- Extensive experiments demonstrate that DFML consistently outperforms state-of-the-art baselines in terms of convergence speed and global accuracy under various conditions.

Weaknesses
- The impact of different network topologies on the performance of DFML is not investigated. The star topology is symmetric and has good stability. I think you should evaluate DFML on more Irregular topologies.
- The experiments do not cover a wide range of real-world datasets and applications, which could help validate the generalizability of the proposed framework.
- The impact of the number of mutual learning epochs K on the performance of DFML is not thoroughly investigated. The paper mentions that increasing K contributes to faster convergence but does not provide a detailed analysis of the trade-off between computational cost and performance improvement.
- Although the framework does not rely on additional data, the privacy risks associated with sharing model parameters and the potential for privacy leaks during the mutual learning process are not addressed. Nowadays, there exsit some powful model inversion attack methods that may cause privacy leakage.

---

> ### Author Response · Authors · 2024-06-17
>
> Dear Reviewer TnFy,
>
> We appreciate your time in reviewing our work and the insightful comments you provided, which have greatly helped us refine and enhance our work. In response to your feedback, we have made revisions to the manuscript, with the modifications indicated in dark red. Below is our detailed response:
>
> 1) Investigate the impact of different network topologies on the performance of DFML.
>
> Response: We introduced a new topology where clients are segregated into two distinct groups. Each group forms a mesh topology internally, while the groups themselves are interconnected via a single link. The first group comprises clients with odd indices [1, 3, 5, ..., N-1], whereas the second group includes clients with even indices [2, 4, 6, ..., N]. The clients from each group that bridge the connection between the two are the ones with "median" address within their respective groups. Figure 23 illustrates this new topology. Our experiments encompass both restrictive and nonrestrictive heterogeneous models applied to two datasets. Results, presented in Tables 11 and 12, demonstrate that even with this new topology, DFML consistently outperforms the established baselines. Section A.4.9 has been included to provide a detailed description of the new topology and the conducted experiments.
>
> 2) Conduct an ablation study to evaluate the individual contributions of the key components of DFML, such as mutual learning, cyclical variation of alpha, and the use of peak models. The impact of the number of mutual learning epochs K on the performance of DFML is not thoroughly investigated. The paper mentions that increasing K contributes to faster convergence but does not provide a detailed analysis of the trade-off between computational cost and performance improvement.
>
> Response: We have added Section A.4.8 to elaborate on the impact of mutual learning compared to vanilla knowledge transfer. Figure 21 illustrates that mutual learning significantly enhances performance. In Section A.4.9, we delve into the influence of increasing mutual learning epochs, denoted as K, with results presented in Figures 22 and 23. Our analysis also covers the effect of K epochs with respect to commutation costs. Results show that higher K epochs improve convergence speed but come at the expense of increased computational costs. For instance, increasing K epochs from 10 to 20 leads to a significant rise in computational requirements  with marginal gains in convergence speed. Conversely, reducing K below 10 reduces computation cost but noticeably slows down convergence. This is why we used K=10 in our experiments. Additionally, Section 5.1 discusses the impact of using peak models, detailed in Figure 8, and Section 5.2 explores the effect of cyclical alpha, as demonstrated in Figure 7, with numerical outcomes provided in Table 9.
>
> 3) Expand the experiments to cover a wider range of real-world datasets and applications.
>
> Response: Three additional datasets, namely Caltech101, Oxford Pets, and Stanford Cars, have been incorporated into our evaluation. Please refer to Tables 5 and 7 for detailed results pertaining to these datasets.
>
> 4) Include a theoretical analysis of the convergence properties and limitations of DFML.
>
> Response: A limitations section (Section 5) has been to the manuscript. One notable limitation of DFML is its computational expense. Additionally, conducting a theoretical analysis on the convergence properties of DFML is beyond the scope of this paper and has been addressed in the limitation section. We acknowledge the significance of this analysis and plan to explore it in future research. Our ongoing and future work will focus on rigorously establishing the convergence behavior of DFML to offer a thorough understanding of its performance and reliability.

---

### Decision · Action_Editor_hVSN · 2024-08-12

**Recommendation:** Accept as is

**Comment:**

The paper provides a new idea for handling heterogeneity in decentralized federated learning systems and proposes a new method of knowledge distillation for clients.

Before rebuttal, reviewers raised several concerns regarding the extensive computational demands of the method, potential biases inherent in the model, the need for more rigorous ablation studies, and a perceived lack of sufficient empirical evidence supporting the method's efficacy. After rebuttal, reviewers acknowledged that major concerns have been addressed.

The limitation of this paper acknowledged by most reviewers is its demand on computational cost, which should not be a major concern for rejecting this paper. The authors have also clearly acknowledged these limitations in their manuscript.

**Audience:**

The paper's approach to tackling heterogeneity in decentralized environments could stimulate further research and discussion. The topic is of current interest within the TMLR audience, particularly for those focusing on decentralized systems and federated learning. The paper's approach to tackling heterogeneity in decentralized environments could stimulate further research and discussion.

**Claims And Evidence:**

The submission introduces an innovative method in decentralized federated learning that addresses data and model heterogeneity through a knowledge distillation process on clients. During rebuttal, the authors have conducted additional ablation studies, provided more empirical evidence to support the performance of their method and acknowledged the extensive computational demand for using their method. Reviewers have recognized that the major claims are now well-supported by the evidence.